# Validation Free and Replication Robust Volume-based Data Valuation

**Xinyi Xu**[†§*], **Zhaoxuan Wu**[‡¶*], **Chuan Sheng Foo**[§], **Bryan Kian Hsiang Low**[†]

Dept. of Computer Science, National University of Singapore, Republic of Singapore[†]
Institute of Data Science, National University of Singapore, Republic of Singapore[‡]
Integrative Sciences and Engineering Programme, NUSGS, Republic of Singapore[¶]
Institute for Infocomm Research, A*STAR, Republic of Singapore[§]
{xuxinyi,lowkh}@comp.nus.edu.sg[†]
wu.zhaoxuan@u.nus.edu[‡¶]
foo_chuan_sheng@i2r.a-star.edu.sg[§]

## Abstract

Data valuation arises as a non-trivial challenge in real-world use cases such as collaborative machine learning, federated learning, trusted data sharing, data marketplaces. The value of data is often associated with the learning performance (e.g., validation accuracy) of a model trained on the data, which introduces a close coupling between data valuation and validation. However, a validation set may not be available in practice and it can be challenging for the data providers to reach an agreement on the choice of the validation set. Another practical issue is that of data replication: Given the value of some data points, a dishonest data provider may replicate these data points to exploit the valuation for a larger reward/payment. We observe that the diversity of the data points is an inherent property of a dataset that is independent of validation. We formalize diversity via the *volume* of the data matrix (i.e., determinant of its left Gram), which allows us to establish a formal connection between the diversity of data and learning performance without requiring validation. Furthermore, we propose a *robust volume* measure with a theoretical guarantee on the replication robustness by following the intuition that copying the same data points does not increase the diversity of data. We perform extensive experiments to demonstrate its consistency in valuation and practical advantages over existing baselines and show that our method is model- and task-agnostic and can be flexibly adapted to handle various neural networks.

## 1 Introduction

Data is increasingly recognized as a valuable resource [19], so we need a principled measure of its worth. A suitable data valuation has wide-ranging applications such as fairly compensating clinical trial researchers for their collected data [12, 16, 25], fostering collaborative machine learning and federated learning among industrial organizations [35, 36, 39], encouraging trusted data sharing and building data marketplaces [7, 30, 32, 37], among others.

A popular viewpoint is that the value of data should correlate with the learning performance of a model trained on the data [14, 18], which enforces a close coupling between data valuation and validation. However, a validation set may not always be available in practice [35]. Also, as different choices of the validation set can lead to different data valuations, it is challenging for the data providers to agree on the choice of such a validation set [35]. Since valuation is coupled with validation, if the

---

*Equal contribution.

validation set is not sufficiently representative of the distribution of test queries in a learning task, the resulting valuation may not be as accurate/useful [40]. We adopt a different perspective: The value of data should be related to its intrinsic properties and valuation can be decoupled from validation by considering the inherent diversity of the data. Intuitively, a more diverse collection of data points corresponds to a higher-quality dataset and thus yields a larger value. This perspective circumvents the above practical limitations and allows our valuation method to be model- and task-agnostic. We formalize diversity via the *volume* of the data matrix (i.e., determinant of its left Gram).

Data replication is another practical issue in data valuation due to the digital nature and anonymous setting of data marketplaces [15]. Supposing a dataset has some value and a data provider instead offers one containing two copies of every data point in this dataset, is this "new" dataset twice as valuable as the original one? Intuitively, the answer should be no as replication adds no new data and so does not increase diversity. We formalize this intuition by constructing a compressed version of the original data to assign little value to replicated data and still preserve its inherent diversity, hence guaranteeing replication robustness.

We provide theoretical justifications for formalizing diversity via volume: Firstly, diversity should be non-negative and monotonic [14, 18, 35, 38] and volume satisfies both properties. Secondly, a greater diversity should lead to a better learning performance [23]: We formally show that a larger volume generally leads to a better performance using the *ordinary least squares* (OLS) framework and our method can be flexibly adapted to handle more complex machine learning models (i.e., various neural networks) in our experiments. Specifically, data with a larger volume can lead to a more accurate pseudo-inverse (i.e., a key component of the least squares solution) and a smaller mean squared error.

To ensure replication robustness, we find that the marginal increase in value from replication must diminish to zero. Otherwise, a data provider can exploit this valuation by making infinite copies of the data to achieve infinite value. We thus formalize the notion of replication robustness via the asymptotic value attainable through replication. Unfortunately, the conventional definition of volume does not have this property. So, we propose a *robust volume* (RV) measure by constructing a compressed version of the original data that groups similar data via discretized cubes of the input feature space and represents those in each cube via a statistic. The RV measure offers practitioners the flexibility to trade off between diversity representation and replication robustness via the cube's width. We perform extensive experiments on synthetic and real-world datasets to demonstrate that our method produces consistent valuations with existing methods while making fewer assumptions.

The specific contributions of our work here include:

- Formalizing a measure of data diversity via the volume of data (Sec. 2) and justifying the suitability of volume for data valuation both theoretically (Sec. 3) and empirically (Sec. 5);
- Formalizing the notion of replication robustness and designing a data valuation method based on the *robust volume* (RV) measure with a theoretical guarantee on replication robustness (Sec. 4);
- Performing extensive empirical comparisons with baselines to demonstrate that our method is consistent in valuation without validation, replication robust, and can be flexibly adapted to handle complex machine learning models such as various neural networks (Sec. 5).

## 2 Problem Setting and Notations

Consider two data submatrices $\mathbf{X}_S$ and $\mathbf{X}_{S'}$ to be valued that contain $s$ and $s'$ rows of $d$-dimensional input feature vectors, respectively. Let $\mathbf{P}_S := [\mathbf{X}_S^\top \ \mathbf{0}]^\top \in \mathbb{R}^{n \times d}$ be the zero-padded version of $\mathbf{X}_S \in \mathbb{R}^{s \times d}$. We concatenate the data submatrices along the rows to form the full data matrix $\mathbf{X} \in \mathbb{R}^{n \times d}$, i.e., $\mathbf{X} := [\mathbf{X}_S^\top \ \mathbf{X}_{S'}^\top]^\top$ and $n = s + s'$. We denote the corresponding observed labels/responses as $\mathbf{y} := [\mathbf{y}_S^\top \ \mathbf{y}_{S'}^\top]^\top \in \mathbb{R}^{n \times 1}$. The least squares solution from OLS is $\mathbf{w} := \mathbf{X}^+ \mathbf{y} = \arg\min_{\boldsymbol{\beta}} \|\mathbf{y} - \mathbf{X}\boldsymbol{\beta}\|^2$ where $\mathbf{X}^+ := (\mathbf{X}^\top \mathbf{X})^{-1} \mathbf{X}^\top$ is the pseudo-inverse of $\mathbf{X}$. Similarly, we denote $\mathbf{X}_S^+$ as the pseudo-inverse of $\mathbf{X}_S$ and $\mathbf{w}_S := \mathbf{X}_S^+ \mathbf{y}_S$. To ease notations, let $V := \text{Vol}(\mathbf{X})$ and $V_S := \text{Vol}(\mathbf{X}_S)$ where $\text{Vol}()$ is defined below. Let $|\mathbf{A}|$ denote the determinant of a square matrix $\mathbf{A}$. The left Gram matrix of $\mathbf{X}$ is $\mathbf{G} := \mathbf{X}^\top \mathbf{X} \in \mathbb{R}^{d \times d}$, so for data submatrix $\mathbf{X}_S$, $\mathbf{G}_S := \mathbf{X}_S^\top \mathbf{X}_S \in \mathbb{R}^{d \times d}$.

**Definition 1 (Volume).** *For a full-rank $\mathbf{X} \in \mathbb{R}^{n \times d}$ with $n \geq d$, $\text{Vol}(\mathbf{X}) := \sqrt{|(\mathbf{X}^\top \mathbf{X})|} = \sqrt{|\mathbf{G}|}$.*

We adopt the above definition of volume for several reasons: (a) Often, the input feature space of the data is pre-determined and fixed due to the data collection process. But, new data can stream in and

so, $n$ can grow indefinitely while $d$ remains fixed [9, 10]. (b) By leveraging the formal connection between volume and learning performance (Sec. 3), we can design a validation free volume-based data valuation to assign a larger value to data leading to a better learning performance. (c) This affords an intuitive interpretation between volume and diversity: Adding a data point to a dataset can increase the diversity/volume depending on the data points already in the dataset (Lemma 1).

We restrict our discussion to full-rank matrices $\mathbf{X}$, $\mathbf{X}_S$, and $\mathbf{X}_{S'}$ since otherwise we can adopt the Gram-Schmidt process to remove the linearly dependent columns [9, 10]. In practice, we perform pre-processing such as principal component analysis to reduce the dimension of the input feature space to ensure that this assumption is satisfied. This assumption is to ensure that there are no redundant features, namely, features that can be exactly reconstructed using other features. For instance, if a dataset already contains monthly salaries, then an annual salary would be redundant.

## 3 Larger Volume Entails Better Learning Performance

The value of a data (sub)matrix depends on the learning performance trained on it [14, 18] which, we will show, depends on its volume. Simply put, the larger the volume, the better the learning performance. In this section, we will formalize this claim through the *ordinary least squares* (OLS) framework. In particular, we will investigate two metrics for learning performance: (a) the quality of the pseudo-inverse represented by $\text{bias}_S := \left\| \mathbf{P}_S^+ - \mathbf{X}^+ \right\|$ because estimating $\mathbf{X}^+$ accurately is important to achieving small *mean squared error* (MSE) [9] and where $\mathbf{P}_S^+ := (\mathbf{X}_S^\top \mathbf{X}_S)^{-1} \mathbf{P}_S^\top$, and (b) the MSE denoted as $L(\mathbf{w}_S) := \left\| \mathbf{y} - \mathbf{X} \mathbf{w}_S \right\|^2$.

### 3.1 Larger Volume Entails Smaller Bias

In regression problems, the closed-form optimal solution is constructed via $\mathbf{X}^+$ computed using $\mathbf{X}$. So, the bias of $\mathbf{P}_S^+$ from $\mathbf{X}^+$ indirectly determines the value of $\mathbf{X}_S$ [9], i.e., a smaller bias means a larger value. We show in Proposition 1 below that 'a larger volume means a smaller bias' always holds for $d = 1$. For $d > 1$, it requires additional assumptions which are mostly satisfied via empirical verification (Fig. 1).

**Proposition 1** (**Volume vs. Bias for** $d = 1$). *For non-zero* $\mathbf{X}_S, \mathbf{X}_{S'}$ *of* $\mathbf{X} \in \mathbb{R}^{n \times 1}$, $V_S \geq V_{S'} \iff \text{bias}_S - \text{bias}_{S'} \leq 0$.

The above result can be generalized to $M > 2$ non-zero data submatrices: Let $\mathbf{X} := [\mathbf{X}_{S_1}^\top \ \mathbf{X}_{S_2}^\top \ \cdots \ \mathbf{X}_{S_M}^\top]^\top$ and w.l.o.g., suppose that $V_{S_1} \geq V_{S_2} \geq \ldots \geq V_{S_M}$. Then, $\text{bias}_{S_1} \leq \text{bias}_{S_2} \leq \ldots \leq \text{bias}_{S_M}$. For $d > 1$, counterexamples exist (see Fig. 1), so we instead compare $\text{bias}_S$ and $\text{bias}_{S'}$ in the next result:

**Proposition 2** (**Volume vs. Bias in General**). *For full-rank* $\mathbf{X}_S, \mathbf{X}_{S'}$ *of* $\mathbf{X} \in \mathbb{R}^{n \times d}$,

$$\text{bias}_S^2 - \text{bias}_{S'}^2 = \frac{1}{V_S^4} \left\| \mathbf{Q}_S \mathbf{X}_S^\top \right\|^2 - \frac{1}{V_{S'}^4} \left\| \mathbf{Q}_{S'} \mathbf{X}_{S'}^\top \right\|^2 + 2 \left\langle \frac{1}{V^2} \mathbf{Q} \mathbf{X}^\top, \frac{1}{V_{S'}^2} \mathbf{Q}_{S'} \mathbf{P}_{S'}^\top - \frac{1}{V_S^2} \mathbf{Q}_S \mathbf{P}_S^\top \right\rangle$$

*where* $\mathbf{Q} := \sum_{l=1}^k (\lambda_l \sigma_l)^{-1} \prod_{j=1, j \neq l}^k (\mathbf{G} - \lambda_j \mathbf{I})$, $\{\lambda_l\}_{l=1}^k$ *denotes the* $k$ *unique eigenvalues of the left Gram matrix* $\mathbf{G}$ *of* $\mathbf{X}$, $\mathbf{Q}_S, \mathbf{Q}_{S'}$ *are similarly defined w.r.t.* $\mathbf{G}_S, \mathbf{G}_{S'}$, $\mathbf{P}_S$ *and* $\mathbf{P}_{S'}$ *are, respectively, zero-padded versions of* $\mathbf{X}_S$ *and* $\mathbf{X}_{S'}$, *and* $\sigma_l := \sum_{g=1}^k (-1)^{g+1} \lambda_l^{k-g} [\sum_{\mathcal{H} \subseteq \{1,\ldots,k\} \setminus \{l\}, |\mathcal{H}| = g-1} (\prod_{h \in \{1,\ldots,k\} \setminus \mathcal{H}} \lambda_h^{-1})]$.

The proof of Proposition 1 (Appendix A.1) relies on a key observation that for $d = 1$, the left Gram matrix is a number and the rest of the proof follows. However, it cannot be generalized to that for $d > 1$, so we resort to a different proof technique. The proof of Proposition 2 requires Lemma 2 in Appendix A.1 which establishes a formal connection between volume and $\mathbf{G}^{-1}$ using the Sylvester's formula. To obtain $V_S \geq V_{S'} \implies \text{bias}_S \leq \text{bias}_{S'}$, there are two cases requiring different additional assumptions: (A) $V_S \gg V_{S'}$, and (B) $\left\| \mathbf{Q}_S \mathbf{X}_S^\top \right\| \approx \left\| \mathbf{Q}_{S'} \mathbf{X}_{S'}^\top \right\|$ and $V \gg \max(V_S, V_{S'})$. Case A is intuitive: $V_S \gg V_{S'}$ means $\mathbf{X}_S$ is much "larger" in volume than $\mathbf{X}_{S'}$, so $\text{bias}_S$ is smaller. Case B is when $\mathbf{X}_S$ and $\mathbf{X}_{S'}$ are similar (e.g., when they are sampled from the same data distribution). The intuition is that the first difference term will be relatively large in magnitude (so, its sign will dominate the overall expression), while the second inner product term will be relatively small in magnitude. This is because the first difference term involves $1/V_S^4$ and $1/V_{S'}^4$ but the second inner

product term involves $1/(V^2 \times V_S^2)$ and $1/(V^2 \times V_{S'}^2)$, and we show $V \gg \max(V_S, V_{S'})$ (Lemma 3 in Appendix A.1). Subsequently, $\|\mathbf{Q}_S \mathbf{X}_S^\top\| \approx \|\mathbf{Q}_{S'} \mathbf{X}_{S'}^\top\|$ and $V_S \geq V_{S'}$ suggest that the first difference term (and thus the overall expression) is likely negative. We empirically verify in Fig. 1 that $V_S \geq V_{S'} \implies \text{bias}_S \leq \text{bias}_{S'}$ holds for more than $80\%$ of times.

## 3.2 Larger Volume Entails Smaller MSE

In Proposition 3 (see proof in Appendix A.2) below, we will show a similar result (to Proposition 1) theoretically analyzing the connection between volume and MSE when $d = 1$, which may be surprising since $\text{Vol}()$ (Definition 1) does not consider $\mathbf{y}$ at all and can yet determine which data submatrix offers better predictions on the rest of the (unobserved) data. Unfortunately, such a result does not directly generalize to $d > 1$ or beyond two submatrices. Nevertheless, we will analyze the effect of volume on the learning performance (i.e., MSE) in general.

**Proposition 3** (**Volume vs. MSE for** $d = 1$). *For non-zero* $\mathbf{X}_S, \mathbf{X}_{S'}$ *of* $\mathbf{X} \in \mathbb{R}^{n \times 1}$, $V_S \geq V_{S'} \iff L(\mathbf{w}_S) - L(\mathbf{w}_{S'}) \leq 0$.

Unfortunately, the above result does not generalize to $d > 1$. For full-rank $\mathbf{X}_S, \mathbf{X}_{S'}$ of $\mathbf{X} \in \mathbb{R}^{n \times d}$, we have derived in Appendix A.2 that

$$L(\mathbf{w}_S) - L(\mathbf{w}_{S'}) = \langle \mathbf{w}_S - \mathbf{w}_{S'}, (\mathbf{X}_S^\top \mathbf{X}_S + \mathbf{X}_{S'}^\top \mathbf{X}_{S'})(\mathbf{w}_S + \mathbf{w}_{S'}) - 2\mathbf{X}^\top \mathbf{y} \rangle \tag{1}$$

and also shown in Appendix A.2 that since $L(\mathbf{w}_S) - L(\mathbf{w}_{S'})$ explicitly depends on $\mathbf{y}$ (1) and $\text{Vol}()$ does not include $\mathbf{y}$ at all, it is possible to adversarially construct $\mathbf{y}$ s.t. $L(\mathbf{w}_S) - L(\mathbf{w}_{S'}) > 0$ or $L(\mathbf{w}_S) - L(\mathbf{w}_{S'}) < 0$ for some fixed $\mathbf{X}_S, \mathbf{X}_{S'}$.

The adversarial cases notwithstanding, volume is regarded as a good surrogate measure of the quality of data applied to active learning and matrix subsampling with theoretical performance guarantees [11, 28]. Similarly, we can adopt the perspective that $\text{Vol}()$ is a measure of the diversity in the input features [23], which provides an intuitive interpretation for Proposition 3: A more diverse dataset with a larger volume gives a better learning performance (i.e., smaller MSE). We will show in Sec. 5.2 that not requiring labels/responses can be an advantage in practice if the labels/responses are noisy/corrupted or there is a distributional difference between the validation and test sets.

We conclude Sec. 3 by empirically verifying whether the additional assumptions described in the last paragraph of Sec. 3.1 are satisfied by checking the percentage of times that $V_S \geq V_{S'} \implies \text{bias}_S - \text{bias}_{S'} \leq 0$ holds. To elaborate, we randomly and identically sample equal-sized $\mathbf{X}_S, \mathbf{X}_{S'}$ over 500 independent trials and compute the percentage of times that a larger volume leads to better learning performance (vertical axis) against the size of $\mathbf{X}_S, \mathbf{X}_{S'}$ (horizontal axis). We consider sampling $\mathbf{X}_S, \mathbf{X}_{S'}$ from either a uniform or normal distribution of varying dimensions: In Fig. 1, for example, '$\mathcal{N}\ d = 1$' denotes $\mathbf{X}_S, \mathbf{X}_{S'}$ being sampled from 1-dimensional standard normal distribution. For MSE, the response $y$ of a data point $\mathbf{x}$ is calculated from $y = \sin(\langle \mathbf{w}^*, \mathbf{x} \rangle)$ where the true parameters $\mathbf{w}^*$ are randomly sampled from $U(0, 2)^d$. Fig. 1 (left) shows that a larger volume leads to a smaller bias for more than $80\%$ of times, thus verifying that the additional assumptions in Sec. 3.1 are satisfied. Fig. 1 (right) shows that a larger volume leads to a smaller MSE for more than $50\%$ of times for $d \leq 10$, which is consistent with the above implications from (1).

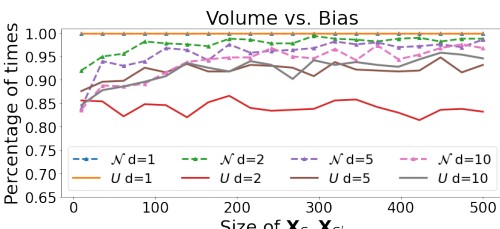 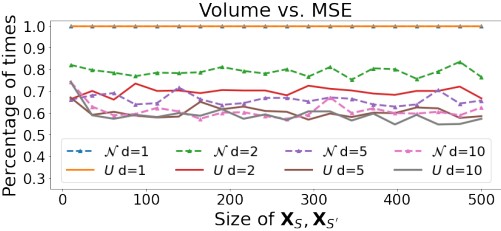

Figure 1: Volume vs. bias (left) and volume vs. MSE (right) for both identically sampled, equal-sized datasets $\mathbf{X}_S, \mathbf{X}_{S'}$ from either a uniform $U(0, 1)^d$ or normal $\mathcal{N}(0, 1)^d$ distribution. The vertical axis shows the percentage of times over 500 independent trials that the dataset with a larger volume leads to a better learning performance (i.e., smaller bias or MSE).

# 4 Robustifying Volume-based Data Valuation

As a larger volume can entail a better learning performance (Sec. 3), we consider a volume-based data valuation method. Unfortunately, volume (Definition 1) is *not* robust to replication via direct data copying. Hence, we will introduce a modified volume measure that can trade off a more refined representation of diversity for greater robustness to replication.

## 4.1 First Attempt of Volume-based Data Valuation

Directly using $\mathrm{Vol}(\mathbf{X})$ as a valuation of $\mathbf{X}$ satisfies both non-negativity and monotonicity which follow directly from Definition 1 and the matrix determinant lemma, respectively:

**Proposition 4 (Non-negativity and Monotonicity of** $\mathrm{Vol}()$**).** *For full-rank* $\mathbf{X} \in \mathbb{R}^{n \times d}$, $\mathrm{Vol}(\mathbf{X}) \geq 0$ *and* $\mathrm{Vol}([\mathbf{X}^\top \ \mathbf{x}^\top]^\top) \geq \mathrm{Vol}(\mathbf{X})$ *where* $\mathbf{x} \in \mathbb{R}^{1 \times d}$ *is a new data point.*

The properties of $\mathrm{Vol}()$ in Proposition 4 imply that a bigger-sized $\mathbf{X}$ (i.e., more data) should yield a larger value [14, 18, 35]. However, $\mathrm{Vol}()$ is unbounded and has a multiplicative scaling factor w.r.t. replication. The implication is that a data provider can arbitrarily "inflate" the volume or value of data by replicating the data infinitely, as shown in the following result (see proof in Appendix A.3):

**Lemma 1 (Unbounded Multiplicative Scaling of** $\mathrm{Vol}(\mathbf{X})$ **from Replication).** *For full-rank* $\mathbf{X} \in \mathbb{R}^{n \times d}$, *let* $\mathbf{x}_q \in \mathbb{R}^{1 \times d}$ *be a data point replicated for* $m \geq 1$ *times and* $\mathbf{X}_{\mathrm{rep}} := [\mathbf{X}^\top \ \mathbf{x}_q^\top \ \ldots \ \mathbf{x}_q^\top]^\top \in \mathbb{R}^{(n+m) \times d}$. *Then,* $\mathrm{Vol}(\mathbf{X}_{\mathrm{rep}}) = \mathrm{Vol}(\mathbf{X}) \times (1 + m \times \mathbf{x}_q (\mathbf{X}^\top \mathbf{X})^{-1} \mathbf{x}_q^\top)^{1/2}$.

**Replication robustness defined via inflation.** We define a measure of *inflation* as the ratio $\nu(\mathrm{replicate}(\mathbf{X}, c))/\nu(\mathbf{X})$ where $\nu()$ is a data valuation function (e.g., $\mathrm{Vol}()$) mapping a data matrix to a real value, the function $\mathrm{replicate}(\mathbf{X}; c)$ directly copies the data in $\mathbf{X}$ and appends them back to $\mathbf{X}$ to output $\mathbf{X}_{\mathrm{rep}} \in \mathbb{R}^{(nc) \times d}$, and the *replication factor* $c$ denotes the amount of replication. One way of replication is to copy the entire $\mathbf{X}$ for $c$ times. Another way is to copy some data submatrix for a certain number of times s.t. $\mathbf{X}_{\mathrm{rep}} \in \mathbb{R}^{(nc) \times d}$. We consider the second way because replicating different data increases the value differently (Lemma 1). We define below a measure of replication robustness to formalize the intuition that greater robustness should guarantee smaller inflation:

**Definition 2 (Replication Robustness of Data Valuation** $\nu()$**).** *Define replication robustness of* $\nu()$ *as* $\gamma_\nu := \nu(\mathbf{X})/(\sup_{c \geq 1} \nu(\mathbf{X}_{\mathrm{rep}}))$ *where* $\mathbf{X}_{\mathrm{rep}} := \mathrm{replicate}(\mathbf{X}, c) \in \mathbb{R}^{(nc) \times d}$.

The theoretically optimal robustness is $\gamma_\nu = 1$, which implies no additional gain from replicating data, hence discouraging replication completely. In contrast, the worst-case robustness is $\gamma_\nu = 0$, which is the case for any $\nu()$ that strictly monotonically increases with replication and, in particular, $\gamma_{\mathrm{Vol}} = 0$ by applying Lemma 1. As a result, a replication robust data valuation function must have a diminishing marginal value from replication: The additional gain from having more copies of the same data converges asymptotically to $0$ w.r.t. $c$. This aligns with what we observe in practice: Repeatedly adding the same data to a training set does not improve the learning performance indefinitely.

## 4.2 Replication Robust Volume (RV)-Based Data Valuation

We will propose an RV measure by constructing a compressed version of original data matrix $\mathbf{X}$ that groups similar data points via discretized cubes of the input feature space and represents those in each cube via a statistic. The RV measure offers practitioners the flexibility to trade off a more refined diversity representation for greater replication robustness by increasing the cube's width.

**Definition 3 (Replication Robust Volume (RV)).** *Let the* $d$-*dimensional input feature space/domain for* $\mathbf{X}$ *be discretized into a set* $\Psi$ *of* $d$-*cubes of width/discretization coefficient* $\omega$, $\phi_i$ *denote the number of data points in* $d$-*cube* $i \in \Psi$, $\boldsymbol{\mu}_i \in \mathbb{R}^{1 \times d}$ *be a statistic (e.g., mean vector) of the data points in* $d$-*cube* $i$, *and* $\widetilde{\mathbf{X}} := [\boldsymbol{\mu}_i^\top]_{i \in \Psi : \phi_i \neq 0}^\top$ *be a compressed version of* $\mathbf{X}$ *s.t. each row of* $\widetilde{\mathbf{X}}$ *is a statistic* $\boldsymbol{\mu}_i$ *of the data points in non-empty* $d$-*cube* $i$. *The replication robust volume is*

$$\mathrm{RV}(\mathbf{X}; \omega) := \mathrm{Vol}(\widetilde{\mathbf{X}}) \times \prod_{i \in \Psi} \rho_i \tag{2}$$

*where* $\rho_i := \sum_{p=0}^{\phi_i} \alpha^p$ *with hyperparameter* $\alpha \in [0, 1]$ *controlling the degree of robustness.*

In contrast to the unbounded $\mathrm{Vol}()$, we ensure that $\mathrm{RV}(;\omega)$ is bounded by setting $\prod_{i\in\Psi}\rho_i$ to be bounded and convergent w.r.t. the size of the replicated data. Note that $\phi_i = 0 \implies \rho_i = 1$ (i.e., an empty $d$-cube) and $\phi_i > 0 \implies \rho_i > 1$. Before considering any robustness guarantee, we will first show in Proposition 5 (see proof in Appendix A.3) below that RV (Definition 3) preserves the original volume in a relative sense, i.e., the ratio $V_S/V_{S'}$ is preserved. The implication is a similar effect of RV on the learning performance (Sec. 3), as empirically demonstrated in Sec. 5.1.

**Proposition 5 (Bounded Distortion of $\mathrm{RV}(\mathbf{X}_S;\omega)/\mathrm{RV}(\mathbf{X}_{S'};\omega)$).** *Define distortion* $\delta(\omega) :=$ $[\mathrm{RV}(\mathbf{X}_S;\omega)/\mathrm{RV}(\mathbf{X}_{S'};\omega)]/[\mathrm{Vol}(\mathbf{X}_S)/\mathrm{Vol}(\mathbf{X}_{S'})]$. *Then,* $(\exp(\beta^{-1}))^{-1} \le \delta(\omega) \le \exp(\beta^{-1})$ *for any* $\omega > 0$ *where* $\beta = 1/(\alpha n)$. *For example,* $\beta = 10$ *bounds* $\delta(\omega) \in [0.905, 1.105]$ *approximately.*

**Near-optimal robustness by upper-bounding inflation.** We have previously defined robustness (Definition 2) as the maximum attainable inflation via replication. Since $\rho_i$ and inflation are monotonic in $\phi_i$, we consider the asymptotic inflation: $\phi_i \to \infty$. In Definition 3, even when the data in $d$-cube $i$ is replicated infinitely many times, the inflation from this $d$-cube is still upper-bounded by a constant. This can be generalized to all the $d$-cubes as each can be considered independently and there is a constant number of $d$-cubes for a fixed $\mathbf{X}$ and $\omega$.

**Proposition 6 (Robustness $\gamma_{\mathrm{RV}}$).** *For* $\alpha \in [0,1)$, $\gamma_{\mathrm{RV}} \ge (1-\alpha)^{|\Psi|}$ *where, with a slight abuse of notation,* $\Psi$ *denotes the set of non-empty $d$-cubes. For* $\alpha = 1, \gamma_{\mathrm{RV}} = 0$.

Its proof is in Appendix A.3. Recall from Definition 2 that $\gamma_{\mathrm{RV}} = 1$ is optimal robustness. From Proposition 6, reducing $\alpha$ achieves a smaller upper bound on inflation and greater robustness. However, if $\alpha$ is too small, then it may have an undesirable effect: $\mathrm{RV}(\mathbf{X};\omega) < \mathrm{Vol}(\mathbf{X})$ for some $\mathbf{X}$ (with similar data points) from an honest provider without replication. In this case, RV has an over-correcting effect: RV is designed to avoid exploitation of $\mathrm{Vol}()$ due to replication but mistakenly leads to a decrease in the value of an honest dataset. Therefore, $\alpha$ should be set to achieve a certain upper bound on inflation but should not be unnecessarily small; more details are given in Proposition 8 in Appendix A.3. In particular, setting $\alpha = 1/(\beta n)$ guarantees a constant upper bound $\exp(\beta^{-1})$ on the inflation, as proven in Lemma 5 in Appendix A.3. For instance, setting $\beta = 10$ and $\alpha = 1/(\beta n)$ guarantees $\mathrm{RV}(\mathrm{replicate}(\mathbf{X}, c);\omega) \le 110\% \times \mathrm{RV}(\mathbf{X};\omega)$. However, it requires us to know the true $n$ *without* any replication. In practice, as we can only observe the data with replication (if any) [15], we estimate $n$ with the number $|\Psi|$ of rows in $\widetilde{\mathbf{X}}$.

**Trading off diversity representation for replication robustness via $\omega$.** A smaller $\omega$ means that the $d$-cubes are more refined and RV can better represent the original data instead of crudely grouping many data points together and representing them via a statistic. On the other hand, a larger $\omega$ means a less refined diversity representation but greater replication robustness. In the extreme case, a sufficiently large $\omega$ results in grouping all data points together and representing them all using a single statistic, hence foregoing the diversity in data. So, a practitioner should determine the trade-off between diversity representation vs. replication robustness based on the requirements of the real-world use case. The following result (see proof in Appendix A.3) formalizes both extremes of the trade-off:

**Proposition 7 (Reduction to $\mathrm{Vol}()$ vs. Optimal Robustness).** *Set $\omega$ to be s.t. each $d$-cube only contains completely identical data points, and*

1. *set $\rho_i$ to some constant $K_{\widetilde{\mathbf{X}},i}$ for $i \in \Psi$ based on a recursive application of Lemma 1. Then, $\mathrm{RV}(\cdot;\omega) = \mathrm{Vol}()$;*
2. *set $\alpha = 0$. So, $\rho_i = \mathbb{1}(\phi_i \neq 0)$ and name this formulation $\mathrm{RV}_{\mathbb{1}}(\cdot;\omega)$. Then, $\gamma_{\mathrm{RV}_{\mathbb{1}}} = 1$.*

$\mathrm{RV}_{\mathbb{1}}(\cdot;\omega)$ can be seen as reducing all potential replications to one data point. It achieves robustness but loses the density information of each $d$-cube due to the indicator function. Specifically, the true distribution may have different densities at different $d$-cubes, which is reflected via $\phi_i$'s. But, this information is completely lost in $\mathrm{RV}_{\mathbb{1}}(\cdot;\omega)$. In contrast, $\mathrm{Vol}()$ represents all the data indiscriminately, hence sacrificing robustness. Furthermore, while we restrict our consideration of replication to direct copying, it is natural to additionally consider a noisy replication (i.e., adding small random perturbations to copies [15]). Intuitively, $\mathrm{RV}_{\mathbb{1}}(\cdot;\omega)$ is not robust to noisy replication as the replicated data are perturbed. Our preliminary empirical study in Appendices B.2 and B.3 shows that RV is robust to noisy replication if the noise magnitude is small relative to $\omega$. So, a future work is to devise a way to optimize the trade-off between diversity representation and replication robustness via $\omega$. In our work here, we empirically find $\omega = 0.1$ suitable for the case of standardized input features.

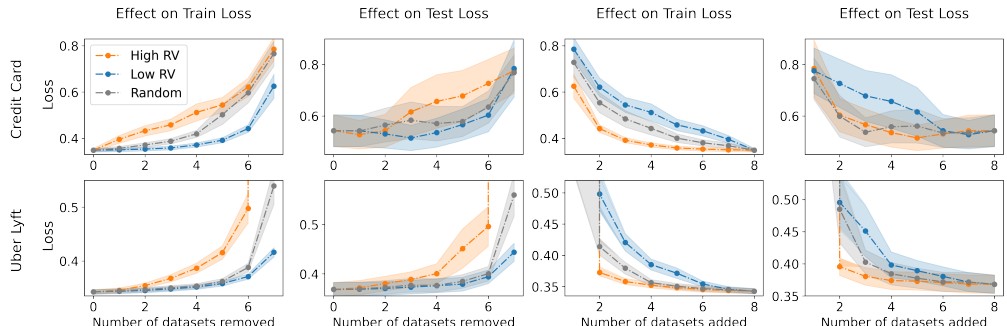

Figure 2: Effect of removing/adding dataset with highest/lowest RV on train/test loss for real-world credit card and Uber Lyft datasets. Plots show the average and standard errors over 50 random trials.

In using standardized input features, we implicitly assume that the input features follow a normal distribution. This makes the data further away from the mean (i.e., statistically rarer) more valuable in learning [11]. We also observe this in Sec. 5.2 where data closer to the mean are valued to be smaller across all baselines and our method. Our work here excludes considerations of outliers as they are not truly representative of the true data distribution.

## 5  Experiments and Discussion

In this section, we will first verify our claim in Sec. 3 that a larger volume leads to a better learning performance and reveal some interesting practical perspectives in Sec. 5.1. Then, in Sec. 5.2, we will show that RV produces results consistent with existing baseline methods and also demonstrate the limitations of these baselines. In particular, RV is model- and task-agnostic while another baseline with an explicit dependence on the validation set is shown to have some deviation in data valuation as the validation set changes. Lastly, in Sec. 5.3, we will verify our robustness guarantee by analyzing its asymptotic behavior under replication. Importantly, our empirical study has gone beyond the OLS framework used for the theoretical analysis in Sec. 3 as our method can be flexibly adapted to handle various neural network architectures on different machine learning tasks including both image classification and natural language processing. All experiments have been run on a server with Intel(R) Xeon(R)@ 2.70GHz processor and 256GB RAM. Our code is publicly available at: `https://github.com/ZhaoxuanWu/VolumeBased-DataValuation`.

### 5.1  Effect of Robust Volume (RV) on Learning Performance

In this subsection, we use RV and volume interchangeably as replication is not considered here and Proposition 5 guarantees that RV preserves the original volume. We consider the setting of sequentially adding/removing the dataset with highest/lowest RV to analyze the effect of RV on the learning performance [14]. We include random selection as a baseline. We simulate 8 data providers to make the results more generalizable. In this experiment, we use two real-world datasets: credit card fraud detection [2] (i.e., transaction amount prediction) and Uber & Lyft [5] (i.e., carpool ride price prediction) which are pre-processed to contain 8 and 12 standardized input features, respectively. Fig. 2 shows the results. Additional results on two other real-world datasets are in Appendix B.4.

It can be observed that adding (resp., removing) a dataset with a larger RV leads to a smaller (resp., larger) train loss, thus verifying Proposition 2 that a larger volume leads to a more accurate pseudo-inverse and smaller train loss in terms of mean squared error. This observation is also consistent with the results on the test loss, albeit with larger standard errors. This confirms (1) that in a higher dimensional input feature space, a larger volume does not immediately guarantee a smaller test loss.

**Interesting practical perspectives.** The results on adding datasets provide justification for a data buyer with a limited budget to spend on datasets with larger RVs first to achieve the best learning performance, thereby resonating with the active learning paradigm [29]. On the other hand, the results on removing datasets sheds light on the following question: If training on all collected datasets is too costly due to memory or time constraints, then which dataset should be removed first without compromising the learning performance much (i.e., the dataset with smallest RV)?

## 5.2 Empirical Comparison of Robust Volume (RV) Shapley Value with Baselines

We will demonstrate that RV without validation gives results consistent with existing baseline methods which may require validation. Then, we will empirically show the limitations of these baselines.

To design principled, fair payments to the data providers, we use (robust) volume as the characteristic function in the commonly used Shapley value to measure the expected marginal contributions of their datasets [14, 18, 35]. Our *robust volume Shapley value* (RVSV) is defined as follows [33]:

$$\text{RVSV}_m \coloneqq (1/M!) \sum_{\mathcal{C} \subseteq \mathcal{M} \setminus \{S_m\}} [|\mathcal{C}|! \times (M - |\mathcal{C}| - 1)!] \times [\text{RV}(\mathbf{X}_{\mathcal{C} \cup \{S_m\}}; \omega) - \text{RV}(\mathbf{X}_{\mathcal{C}}; \omega)] \quad (3)$$

where $\mathcal{M} \coloneqq \{S_1, \ldots, S_M\}$ denotes a set of $M$ data providers/datasets and $\mathbf{X}_{\mathcal{C}}$ denotes a data matrix constructed from concatenating the data matrix $\mathbf{X}_{S_{m'}}$ of every data provider $S_{m'} \in \mathcal{C} \subseteq \mathcal{M}$. Our *volume Shapley value* (VSV) is computed by replacing $\text{RV}(\cdot; \omega)$ in (3) with $\text{Vol}()$. We compare VSV and RVSV with the following baselines: validation loss *leave-one-out* (LOO) value [21, 27], *validation loss Shapley value* (VLSV) [14, 18], and *information gain Shapley value* (IGSV) [35]. We consider the contributions of $M = 3$ data providers/matrices/datasets $\mathbf{X}_{S_1}$, $\mathbf{X}_{S_2}$, and $\mathbf{X}_{S_3}$ [35]. The input features are standardized and we set $\omega = 0.1$. LOO and VLSV use MSE on a validation set.

**Synthetic data from baseline distributions.** We first consider simpler experimental settings on synthetic data drawn from the 6D Hartmann function [24] defined over $[0, 1]^6$ with four baseline data distributions for $\mathbf{X}_{S_1}$, $\mathbf{X}_{S_2}$, and $\mathbf{X}_{S_3}$: (A) *independent and identical distribution* (i.i.d.) where $\mathbf{X}_{S_1}$, $\mathbf{X}_{S_2}$, and $\mathbf{X}_{S_3}$ contain 200 i.i.d. samples each; (B) ascending dataset size where $\mathbf{X}_{S_1}$, $\mathbf{X}_{S_2}$, and $\mathbf{X}_{S_3}$ contain 20, 50, and 200 i.i.d. samples, respectively; (C) disjoint input domains where $\mathbf{X}_{S_1}$, $\mathbf{X}_{S_2}$, and $\mathbf{X}_{S_3}$ are sampled from the input domains of $[0, 1/3]^6$, $[1/3, 2/3]^6$, and $[2/3, 1]^6$, respectively; and (D) supersets $\mathbf{X}_{S_1} \subset \mathbf{X}_{S_2} \subset \mathbf{X}_{S_3}$ with the respective sizes 200, 400, and 600 where $\mathbf{X}_{S_2}$ (resp., $\mathbf{X}_{S_3}$) has 200 i.i.d. data samples in addition to $\mathbf{X}_{S_1}$ (resp., $\mathbf{X}_{S_2}$).

The results in Fig. 3 show that both VSV and RVSV are generally consistent with IGSV. For (B) ascending dataset size, VSV, RVSV, and IGSV increase from $\mathbf{X}_{S_1}$ to $\mathbf{X}_{S_3}$, while VLSV surprisingly values the contributions of $\mathbf{X}_{S_1}$, $\mathbf{X}_{S_2}$, and $\mathbf{X}_{S_3}$ to be nearly equal; the latter may be due to VLSV's sensitivity to the definition of the value $\nu(\emptyset)$ of an empty dataset/matrix $\emptyset$ when calculating the Shapley value. Fig. 4 illustrates that for i.i.d., VLSV is sensitive to the definition of $\nu(\emptyset)$: For example, setting $\nu(\emptyset)$ to 0 [18], 1.06 (by initializing parameters to zeros), and 8.75 (by initializing parameters randomly using $\mathcal{N}(0, 1)$ [14]) yield different VLSVs of 0.346, 0.183, and 0.330 for $\mathbf{X}_{S_1}$, respectively. These conflicting choices of $\nu(\emptyset)$ add to the difficulties of applying VLSV in practice.

Interestingly, under (C) disjoint input domains, all methods unanimously value the contribution of $\mathbf{X}_{S_2}$ to be the lowest despite their input domains to be of the same size, which is due to the standardization of the input features and so offers the following interpretation: The data in the "center" is the most common if we assume the true data distribution follows a normal one. Therefore, the most common data are valued less while the statistically "rarer" data at the two tails of the distribution are valued more. Additional experimental results with this distribution are reported in Appendix B.5. It is counter-intuitive to see that for i.i.d., LOO values the contribution of $\mathbf{X}_{S_1}$ to be 0, which may be due to instability from the calculation of their contributions [8].

**Real-world datasets with different preferences of validation sets.** We use two real-world datasets: UK used car dataset [1] (i.e., car price prediction) and credit card fraud detection dataset [2] (i.e., transaction amount prediction) where there are different preferences of validation sets [35]. For instance, car dealers for different manufacturers such as Audi, Ford, and Toyota may have different preferences over data. So, we construct two different validation sets comprising cars from different manufacturers. Similarly, different financial institutions may differ in their interests of the transaction amounts. For example, smaller banks typically manage and focus on smaller transaction amounts, so we construct two different validation sets comprising large (i.e., > $1000) vs. small transaction amounts. The results in Fig. 5 show that the effect of different preferences of validation sets on LOO is pronounced, as expected. The effect on VLSV is less due to the averaging of marginal contributions. On the other hand, there is no effect on IGSV, VSV, and RVSV as they do not require a validation set.

## 5.3 Replication Robustness

We first perform a simpler experiment to demonstrate the effect of replication and then perform more extensive experiments under more complex settings to show the asymptotic behavior of RVSV and existing baseline methods under replication.

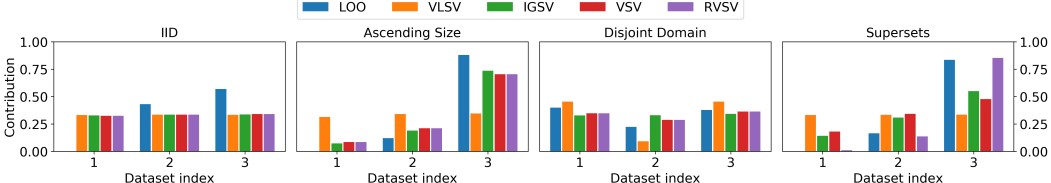

Figure 3: Contributions of $\mathbf{X}_{S_1}$, $\mathbf{X}_{S_2}$, and $\mathbf{X}_{S_3}$ from Hartmann function with baseline data distributions: (A) i.i.d., (B) ascending dataset size, (C) disjoint input domains, and (D) supersets.

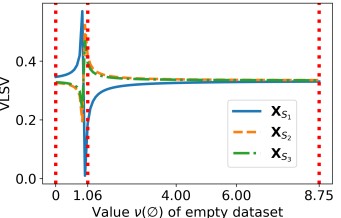

Figure 4: Sensitivity of VLSV to varying $\nu(\emptyset)$ (e.g., 3 red dotted lines).

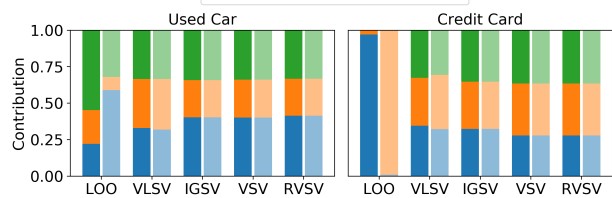

Figure 5: Contributions of $\mathbf{X}_{S_1}$, $\mathbf{X}_{S_2}$, and $\mathbf{X}_{S_3}$ for 2 validation sets distinguished by darker vs. lighter shades.

**Contributions of $\mathbf{X}_{S_1}$, $\mathbf{X}_{S_2}$, and $\mathbf{X}_{S_3}$ under i.i.d. setting.** We perform this experiment on the Trip Advisor hotel reviews dataset [4] (i.e., numerical rating prediction) which contains text reviews data. We utilize the GloVe [31] word embeddings and a bidirectional long short-term memory model with a fully-connected layer of $8$ hidden units. Regression is performed over the $8$-dimensional latent features from this model. Data matrices $\mathbf{X}_{S_1}$, $\mathbf{X}_{S_2}$, and $\mathbf{X}_{S_3}$ follow an i.i.d. partition of the processed data and subsequently, $\mathbf{X}_{S_2}$ and $\mathbf{X}_{S_3}$ are replicated for $2$ and $10$ times, respectively. The results in Fig. 6 show noticeable increases in the contribution of $\mathbf{X}_{S_3}$ for IGSV and VSV, which implies that they are not replication robust. On the other hand, both VLSV and RVSV appear robust.

**Contributions of $\mathbf{X}_{S_1}$, $\mathbf{X}_{S_2}$, and $\mathbf{X}_{S_3}$ under non-i.i.d. settings.** As our replication robustness includes $\sup_c$ (Definition 2), we investigate large replication factors $c$ of up to $100$. Since the previous experiment shows that VLSV is robust, we use it as the baseline for comparison. We additionally consider two non-i.i.d. data distributions extended from the previous setting: *supersets* and *disjoint input domains* for $4$ real-world datasets: California housing price prediction (CaliH) [20], Kings county housing sales prediction (KingH) [3], US census income prediction (USCensus) [6], and age estimation from facial images (FaceA) [41]. We use $60\%$ of data to construct $\mathbf{X}_{S_1}$, $\mathbf{X}_{S_2}$, and $\mathbf{X}_{S_3}$ and the remaining $40\%$ as the validation set for LOO and VLSV. For i.i.d. and supersets, we set $\mathbf{X}_{S_2} = \mathbf{X}_{S_1}$ s.t. $\mathbf{X}_{S_2}$ simulates an honest data provider and we examine the effect of replicating $\mathbf{X}_{S_1}$. For supersets, we vary the proportion of data from $\mathbf{X}_{S_1}$ that is contained in $\mathbf{X}_{S_3}$: If the ratio is $0.1$, then $\mathbf{X}_{S_3}$ contains $10\%$ data from $\mathbf{X}_{S_1}$; if the ratio is $1$, then $\mathbf{X}_{S_1} \subset \mathbf{X}_{S_3}$. For disjoint input domains, we vary how disjoint they are for $\mathbf{X}_{S_1}$, $\mathbf{X}_{S_2}$, and $\mathbf{X}_{S_3}$ via a ratio: $0$ (resp., $1$) means that $\mathbf{X}_{S_1}$, $\mathbf{X}_{S_2}$, and $\mathbf{X}_{S_3}$ have completely disjoint (resp., overlapped) input domains. In other words, with ratio $0$, they do not contain any similar data, while with ratio $1$, they may contain some similar data. Fig. 7 shows results for two datasets with i.i.d. data distribution. For CaliH, we use the latent features from the last layer of a neural network with $2$ fully connected layers of $64$ and $10$ hidden units and

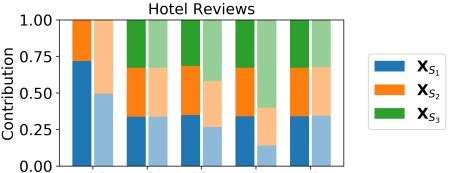

Figure 6: Effect of replication on contributions of $\mathbf{X}_{S_1}$, $\mathbf{X}_{S_2}$, $\mathbf{X}_{S_3}$. Darker (lighter) shade denotes before (after) replication.

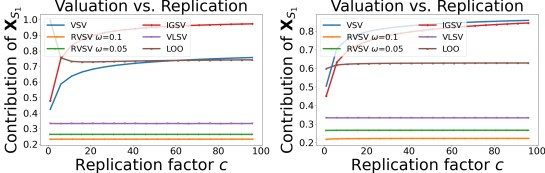

Figure 7: Contribution of the replicated $\mathbf{X}_{S_1}$ with varying replication factors $c$ for CaliH (left) and FaceA (right) datasets.

the *rectified linear unit* (ReLU) as the activation function. Additional details on data distributions, datasets, and models are in Appendix B.6.

Next, we compare the similarity of RVSV and baseline methods to VLSV using similarity measures such as the Pearson correlation coefficient ($r_p$) [18], cosine similarity (cos), and the reciprocal of the $l_2$ norm of the difference [36]. For RVSV, we set $\omega = 0.05$ and $0.1$, which are respectively denoted by RVSV-005 and RVSV-01. Table 1 shows results averaged over varying replication factors $c$ for CaliH; the other results are reported in Appendix B.6. VSV and IGSV are not robust and may be exploited as both increase relatively quickly with replication for $c < 20$ (Fig. 7). Furthermore, our additional experiments on varying hyperparameter choices (Appendix B.7) show that IGSV is sensitive to the choice of hyperparameter whereas RVSV is consistent, even with varying $\omega$. From Fig. 7, RVSV is replication robust. RVSV can also achieve a high degree of similarity to VLSV without requiring validation, as seen in Table 1.

Table 1: Effect of replication on similarity of RVSV and existing baseline methods to VLSV for CaliH dataset. Values in bold indicate the best results.

| Method | i.i.d. | | | disjoint 0 | | | disjoint 1 | | | supersets 0.1 | | | supersets 1 | | |
|---|---|---|---|---|---|---|---|---|---|---|---|---|---|---|---|
| | $r_p$ | cos | $1/l_2$ | $r_p$ | cos | $1/l_2$ | $r_p$ | cos | $1/l_2$ | $r_p$ | cos | $1/l_2$ | $r_p$ | cos | $1/l_2$ |
| LOO | -0.991 | 0.730 | 1.894 | -0.459 | 0.816 | 2.457 | **-0.488** | 0.406 | 0.770 | -0.339 | 0.801 | 2.362 | -0.590 | 0.771 | 2.100 |
| IGSV | -0.903 | 0.637 | 1.591 | 0.640 | 0.639 | 1.583 | -0.763 | 0.636 | 1.589 | -0.893 | 0.636 | 1.580 | -0.716 | 0.653 | 1.687 |
| VSV | -0.886 | 0.787 | 2.493 | 0.644 | 0.784 | 2.415 | -0.780 | 0.775 | 2.335 | -0.892 | 0.779 | 2.389 | -0.660 | 0.813 | 2.696 |
| RVSV-005 | **0.767** | **0.959** | **5.857** | **0.700** | **1.000** | **77.714** | -0.784 | **0.998** | **28.479** | **0.810** | **0.983** | **9.314** | **0.918** | **0.946** | **5.051** |
| RVSV-01 | **0.767** | 0.920 | 4.055 | 0.351 | 0.999 | 47.066 | -0.939 | 0.997 | 20.845 | 0.808 | 0.976 | 7.839 | 0.917 | 0.914 | 3.901 |

# 6 Related Work

Data valuation methods assign a larger value to data that leads to a better learning performance [14, 18, 35, 40]. Existing methods such as leave-one-out approaches [18], the Shapley value-based methods [14, 17], and a reinforcement learning framework [40] require validation. Due to the tight coupling between valuation and validation, these methods may face practical limitations arising from using a validation set (Sec. 1). The work of [35] has proposed an information-theoretic approach to valuing data based on the *information gain* (IG) on the model parameters to avoid the need for validation. However, it has not proven that a larger IG (value) leads to a better learning performance. Our method has this desirable theoretical property without needing validation (Sec. 3). While existing methods demonstrate some effectiveness against replication using carefully selected validation sets [14, 18], our method achieves such a guarantee without needing validation. The work of [15] has considered replication from a different perspective and is thus not directly comparable to our method.

# 7 Conclusion and Future Work

This paper describes a model- and task-agnostic replication robust data valuation method that requires no validation. In particular, we value data based on its inherent diversity formalized as the volume of the data matrix because we have shown in Sec. 3 that a larger volume entails a better learning performance. We have identified that volume is not robust to replication, so we design a data valuation method based on the novel *robust volume* (RV) measure with a theoretical guarantee on replication robustness (Sec. 4). In our experiments (Sec. 5), we have used RV as a characteristic function in the Shapley value and empirical comparison with existing baseline methods verifies its effectiveness in data valuation and its robustness guarantee. Importantly, we have tested on various real-world datasets and our robust volume data valuation method can be flexibly adapted to handle machine learning models more complex than OLS (i.e., various neural networks) to demonstrate its practical applicability. Current works on data pricing may build on our perspective to ease the dependence on the validation set. For future work, we plan to consider more sophisticated replication techniques and investigate how to optimize the trade-off between diversity representation vs. replication robustness.

## Acknowledgments and Disclosure of Funding

This research is supported by the National Research Foundation, Singapore under its AI Singapore Programme (Award No: AISG2-RP-2020-018). Any opinions, findings and conclusions or recom-

mendations expressed in this material are those of the author(s) and do not reflect the views of National Research Foundation, Singapore. Xinyi Xu is supported by the Institute for Infocomm Research of Agency for Science, Technology and Research (A*STAR). The authors thank Fusheng Liu for many interesting discussions.

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
