# A Proofs and Derivations

## A.1 Larger Volume Entails Smaller Bias

***Proof of Proposition 1.*** Let $\mathbf{P}_S$ be the zero-padded version of $\mathbf{X}_S$ such that $\mathbf{P}_S \in \mathbb{R}^{n \times d}$. In this proof, we only consider $d = 1$. Note $\|\mathbf{P}_S\|^2 = \|\mathbf{X}_S\|^2 = |\mathbf{X}_S^\top \mathbf{X}_S| = \mathrm{Vol}(\mathbf{X}_S)^2$ for $d = 1$. Recall the pseudo-inverse $\mathbf{X}^+ := (\mathbf{X}^\top \mathbf{X})^{-1} \mathbf{X}^\top$.

$$
\begin{aligned}
\|\mathbf{X}^+ - \mathbf{P}_S^+\|^2 &= \left\| \frac{1}{\|\mathbf{X}\|^2} \mathbf{X}^\top - \frac{1}{\|\mathbf{X}_S\|^2} \mathbf{P}_S^\top \right\|^2 \\
&= \frac{1}{\|\mathbf{X}\|^4} \|\mathbf{X}^\top\|^2 + \frac{1}{\|\mathbf{X}_S\|^4} \|\mathbf{P}_S^\top\|^2 - \frac{2}{\|\mathbf{X}\|^2 \|\mathbf{X}_S\|^2} \langle \mathbf{X}^\top, \mathbf{P}_S^\top \rangle \\
&= \frac{1}{\|\mathbf{X}\|^2} + \frac{1}{\|\mathbf{X}_S\|^2} - \frac{2}{\|\mathbf{X}\|^2 \|\mathbf{X}_S\|^2} \|\mathbf{X}_S\|^2 \\
&= \frac{1}{\|\mathbf{X}_S\|^2} - \frac{1}{\|\mathbf{X}\|^2} \\
&= \frac{1}{\mathrm{Vol}(\mathbf{X}_S)^2} - \frac{1}{\mathrm{Vol}(\mathbf{X})^2}
\end{aligned}
$$

Since $\mathrm{Vol}(\mathbf{X})$ is constant, larger the $\mathrm{Vol}(\mathbf{X}_S)$, smaller the square of the bias $\|\mathbf{X}^+ - \mathbf{P}_S^+\|^2$. The proof of the proposition is complete. $\qquad\square$

The proof above establishes a direct connection between $\mathrm{bias}_S$ and $1/V_S^2 - 1/V^2$. Therefore, we can extend to $M > 2$ non-zero submatrices such that, for any $\mathbf{X}_S, \mathbf{X}_{S'} \in \{\mathbf{X}_{S_1}, \mathbf{X}_{S_2}, \ldots, \mathbf{X}_{S_M}\}$, Proposition 1 still holds.

***Proof of Proposition 2.*** The proof is relatively straightforward and follows the expansion of the l.h.s and substituting $\mathbf{X}^+$ by $\mathbf{G}^{-1} \mathbf{X}^\top$ and $\mathbf{X}_S^+$ by $\mathbf{G}_S^{-1} \mathbf{P}_S^\top$. Here, $\mathbf{G}_S := \mathbf{X}_S^\top \mathbf{X}_S$ and $\mathbf{P}_S$ is the zero-padded version of $\mathbf{X}_S$. We use the expression and definitions of $\mathbf{Q}, \mathbf{Q}_S, \mathbf{Q}_{S'}$ in Lemma 2:

$$
\begin{aligned}
\mathrm{bias}_S^2 - \mathrm{bias}_{S'}^2 &:= \|\mathbf{X}^+ - \mathbf{P}_S^+\|^2 - \|\mathbf{X}^+ - \mathbf{P}_{S'}^+\|^2 \\
&= \|\mathbf{X}^+\|^2 - 2\langle \mathbf{X}^+, \mathbf{P}_S^+ \rangle + \|\mathbf{P}_S^+\|^2 - \|\mathbf{X}^+\|^2 + 2\langle \mathbf{X}^+, \mathbf{P}_{S'}^+ \rangle - \|\mathbf{P}_{S'}^+\|^2 \\
&= \|\mathbf{P}_S^+\|^2 - \|\mathbf{P}_{S'}^+\|^2 + 2\langle \mathbf{X}^+, \mathbf{P}_{S'}^+ - \mathbf{P}_S^+ \rangle \\
&= \frac{1}{V_S^4} \|\mathbf{Q}_S \mathbf{P}_S^\top\|^2 - \frac{1}{V_{S'}^4} \|\mathbf{Q}_{S'} \mathbf{P}_{S'}^\top\|^2 + 2\langle \frac{1}{V^2} \mathbf{Q} \mathbf{X}^\top, \frac{1}{V_{S'}^2} \mathbf{Q}_{S'} \mathbf{P}_{S'}^\top - \frac{1}{V_S^2} \mathbf{Q}_S \mathbf{P}_S^\top \rangle \\
&= \frac{1}{V_S^4} \|\mathbf{Q}_S \mathbf{X}_S^\top\|^2 - \frac{1}{V_{S'}^4} \|\mathbf{Q}_{S'} \mathbf{X}_{S'}^\top\|^2 + 2\langle \frac{1}{V^2} \mathbf{Q} \mathbf{X}^\top, \frac{1}{V_{S'}^2} \mathbf{Q}_{S'} \mathbf{P}_{S'}^\top - \frac{1}{V_S^2} \mathbf{Q}_S \mathbf{P}_S^\top \rangle.
\end{aligned}
$$

$\qquad\square$

**Theorem 1** (**Sylvester's Matrix Theorem**). *Given a diagonalizable square matrix* $\mathbf{A}$ *and an analytic function* $f(\cdot)$, *we have,*

$$
f(\mathbf{A}) = \sum_{l=1}^{k} f(\lambda_l) \mathbf{A}_l \tag{4}
$$

*where* $\lambda_l$ *is the* $l$-*th distinct eigenvalue of* $\mathbf{A}$ *and* $\mathbf{A}_l$ *is the Frobenius covariant defined as follows,*

$$
\mathbf{A}_l := \prod_{j=1, j \neq l}^{k} \frac{1}{\lambda_l - \lambda_j} (\mathbf{A} - \lambda_j \mathbf{I}).
$$

**Corollary 1.** *Suppose* $f(\mathbf{A}) = \mathbf{A}^{-1}$, *then*

$$
\mathbf{A}^{-1} = f(\mathbf{A}) := \sum_{l=1}^{k} \frac{1}{\lambda_l} \mathbf{A}_l.
$$

**Lemma 2 (Expressing $\mathbf{G}^{-1}$ in $\mathrm{Vol}(\mathbf{X})$).** *With $\mathbf{G} := \mathbf{X}^\top \mathbf{X}$, then its inverse has*

$$\mathbf{G}^{-1} = V^{-2} \sum_{l=1}^{k} (\lambda_l \sigma_l)^{-1} \prod_{j=1, j \neq l}^{k} (\mathbf{G} - \lambda_j \mathbf{I}) \tag{5}$$

*where $\lambda_l$ is the $l$-th distinct eigenvalue of $\mathbf{G}$, and constant $\sigma_l$ is defined as follows,*

$$\sigma_l := \sum_{g=1}^{k} (-1)^{g+1} \lambda_l^{k-g} \left[ \sum_{\mathcal{H} \subseteq \{1,\ldots,k\} \setminus \{l\}, |\mathcal{H}|=g-1} \left( \prod_{h \in \{1,\ldots,k\} \setminus \mathcal{H}} \lambda_h^{-1} \right) \right].$$

*We define $\mathbf{Q} := \sum_{l=1}^{k} (\lambda_l \sigma_l)^{-1} \prod_{j=1, j \neq l}^{k} (\mathbf{G} - \lambda_j \mathbf{I})$ for convenience.*

***Proof of Lemma 2.*** The proof uses a key result, Sylvester's matrix theorem, specifically the corollary for the inverse of a matrix (reproduced above in Corollary 1) and properties of the left Gram matrix $\mathbf{G} := \mathbf{X}^\top \mathbf{X}$ such as invertibility and positive definiteness when $\mathbf{X}$ is full-rank.

First, observe since $\mathbf{G}$ is a real symmetric matrix, it is diagonalizable, hence a direct application of the corollary above gives

$$\mathbf{G}^{-1} = \sum_{l=1}^{k} \frac{1}{\lambda_l} \mathbf{G}_l \tag{6}$$

where $\mathbf{G}_l$ is the Frobenius covariant defined in the above theorem and we consider $\mathbf{G}_l$ on its own,

$$\begin{aligned}
\mathbf{G}_l &:= \prod_{j=1, j \neq l}^{k} \frac{1}{\lambda_l - \lambda_j} (\mathbf{G} - \lambda_j \mathbf{I}) \\
&= \underbrace{\prod_{j=1, j \neq l}^{k} \frac{1}{\lambda_l - \lambda_j}}_{p_l} \times \underbrace{\prod_{j=1, j \neq l}^{k} (\mathbf{G} - \lambda_j \mathbf{I})}_{\mathbf{M}_l}
\end{aligned} \tag{7}$$

Observe the denominator of the expanded $p_l$ is a summation of terms that are products of multiple $\lambda_l$'s and all with coefficient either $1$ or $-1$. To see from a specific and self-contained example: suppose $k = 4 = l$, so

$$\begin{aligned}
p_l &= \frac{1}{\lambda_4 - \lambda_1} \times \frac{1}{\lambda_4 - \lambda_2} \times \frac{1}{\lambda_4 - \lambda_3} \\
&= \frac{1}{\lambda_4^3 - \lambda_4^2 \lambda_1 - \lambda_4^2 \lambda_2 - \lambda_4^2 \lambda_3 + \lambda_4 \lambda_1 \lambda_2 + \lambda_4 \lambda_1 \lambda_3 + \lambda_4 \lambda_2 \lambda_3 - \lambda_1 \lambda_2 \lambda_3}.
\end{aligned}$$

Since it is easier to work with $\frac{1}{p_l}$, we derive the following formula for it by extracting a common factor of $\Lambda := \prod_{i=l}^{k} \lambda_l$ to give

$$\frac{1}{p_l} = \Lambda \sum_{g=1}^{k} (-1)^{g+1} \lambda_l^{k-g} \underbrace{\left[ \sum_{\mathcal{H} \subseteq \{1,\ldots,k\} \setminus \{l\}, |\mathcal{H}|=g-1} \left( \prod_{h \in \{1,\ldots,k\} \setminus \mathcal{H}} \frac{1}{\lambda_h} \right) \right]}_{\sigma_l}. \tag{8}$$

Using the result that determinant of the left Gram matrix is the product of its eigenvalues, we have $|\mathbf{G}| = \Lambda$, and substituting the definition of $\sigma_l$, we rewrite (7) as follows,

$$\mathbf{G}_l = \frac{1}{|\mathbf{G}|} \frac{1}{\sigma_l} \mathbf{M}_l. \tag{9}$$

Recalling $\mathrm{Vol}(\mathbf{X})^2 = |\mathbf{G}|$ and plugging (9) back into (6) gives

$$\mathbf{G}^{-1} = \frac{1}{\mathrm{Vol}(\mathbf{X})^2} \underbrace{\sum_{l=1}^{k} \frac{1}{\lambda_l} \frac{1}{\sigma_l} \mathbf{M}_l}_{\mathbf{Q}}.$$

$\square$

**Examining the additional scenario of case 2) in Proposition 2**. The scenario is where $\mathbf{X}_S, \mathbf{X}_{S'}$ are similar in the sense that they may contain a similar number of rows, and are drawn from the same distribution. We focus on showing $V \gg \max(V_S, V_{S'})$ and assume $\left\| \mathbf{Q}_S \mathbf{X}_S^\top \right\| \approx \left\| \mathbf{Q}_{S'} \mathbf{X}_{S'}^\top \right\|$ (which we verify empirically later by showing Proposition 2 is true most of the time in Fig.1).

Lemma 3 states that $V^2$ is larger than $V_S^2, V_{S'}^2$ by a multiplicative factor which is exponential in the number of rows in $V_{S'}, V_S$. See Fig. 8 for an illustration.

**Lemma 3** ($V$ **vs.** $V_S, V_{S'}$). *Let $V, V_S, V_{S'}$ be the respective volumes of $\mathbf{X}, \mathbf{X}_S, \mathbf{X}_{S'}$ and let $s, s'$ denote the respective number of rows in $\mathbf{X}_S, \mathbf{X}_{S'}$. Assume $\mathbf{X}, \mathbf{X}_S, \mathbf{X}_{S'}$ are all full-rank, we have*

$$V^2 > \max((1 + \xi_{S'})^{s'} V_S^2, (1 + \xi_S)^s V_{S'}^2)$$

*where $\xi_{S'} := \min_{\mathbf{x}_q \in \mathbf{X}_{S'}} \mathbf{x}_q (\mathbf{X}_S^\top \mathbf{X}_S)^{-1} \mathbf{x}_q^\top > 0$ and $\xi_S := \min_{\mathbf{x}_q \in \mathbf{X}_S} \mathbf{x}_q (\mathbf{X}_{S'}^\top \mathbf{X}_{S'})^{-1} \mathbf{x}_q^\top > 0$.*

***Proof of Lemma 3***. This is a constructive proof. We will add rows one by one from $\mathbf{X}_{S'}$ to $\mathbf{X}_S$ to finally construct $\mathbf{X}$. For an arbitrary row $\mathbf{x}_q$ from $\mathbf{X}_{S'}$, we have

$$
\begin{aligned}
|\mathbf{X}_{S \cup \{q\}}^\top \mathbf{X}_{S \cup \{q\}}| &= |\mathbf{X}_S^\top \mathbf{X}_S + \mathbf{x}_q^\top \mathbf{x}_q| \\
&= \underbrace{(1 + \mathbf{x}_q (\mathbf{X}_S^\top \mathbf{X}_S)^{-1} \mathbf{x}_q^\top)}_{\text{coeff}_\emptyset} |\mathbf{X}_S^\top \mathbf{X}_S| \\
&\geq (1 + \xi_{S'}) V_S^2
\end{aligned}
$$

The second equality uses the matrix determinant lemma. We can repeat this addition for every row in $\mathbf{X}_{S'}$. Note after adding $\mathbf{x}_q$, and we want to add a different row $\mathbf{x}_{q'}$, in the second line the new coefficient (having added $\mathbf{x}_q$) is $\text{coeff}_{\{q\}} = (1 + \mathbf{x}_{q'} (\mathbf{X}_{S \cup \{q\}}^\top \mathbf{X}_{S \cup \{q\}})^{-1} \mathbf{x}_{q'}^\top)$ and $\text{coeff}_{\{q\}} \geq \text{coeff}_\emptyset$ because $(\mathbf{X}_{S \cup \{q\}}^\top \mathbf{X}_{S \cup \{q\}})^{-1}$ is positive definite and the previously added row $\mathbf{x}_q$ now makes a non-negative contribution to the sum, therefore $(1 + \xi_{S'})$ is still a valid lower-bound for $\text{coeff}_{\{q\}}$. In addition, obviously $|\mathbf{X}_S^\top \mathbf{X}_S| \leq |\mathbf{X}_{S \cup \{q\}}^\top \mathbf{X}_{S \cup \{q\}}|$ so $V_S^2$ is a valid lower bound. Recursively adding this for $s'$ times gives the desired lower bound of $(1 + \xi_{S'})^{s'} V_S^2$. The result then follows. $\square$

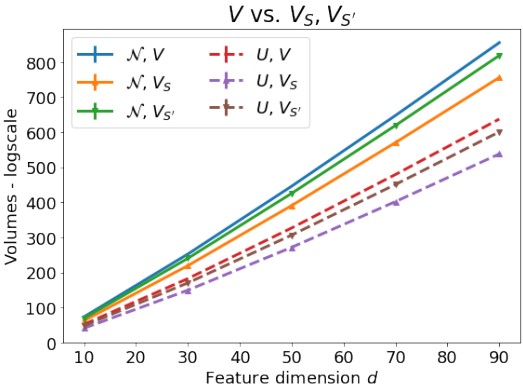

Figure 8: The volume of full matrix $\mathbf{X}$ against the volumes of submatrices $\mathbf{X}_S, \mathbf{X}_{S'}$. $\mathbf{X}_S, \mathbf{X}_{S'}$ are randomly sampled from normal ($\mathcal{N}$, solid lines) and uniform ($U$, dashed lines) distributions and concatenated to form $\mathbf{X}$, with $\mathbf{X}_{S'}$ containing twice the number of rows as in $\mathbf{X}_S$. The results are in log-scale. The volume of the full matrix $\mathbf{X}$ is noticeably larger than $V_S, V_{S'}$ even in log-scale, indicating the actual volume is significantly larger, validating our claim that $V \gg \max(V_S, V_{S'})$.

## A.2 Larger Volume Entails Smaller MSE

***Proof of Proposition 3***. Recall the least squares solution from OLS on training set $(\mathbf{X}, \mathbf{y})$ is $\mathbf{w} = (\mathbf{X}^\top \mathbf{X})^{-1} \mathbf{X}^\top \mathbf{y}$. In the case of $d = 1$, the least squared solution $\mathbf{w}$ is a scalar. We simplify the

notation by letting $w$ and $w'$ be the least squared solutions on training set $(\mathbf{X}_S, \mathbf{y}_S)$ and $(\mathbf{X}_{S'}, \mathbf{y}_{S'})$. Now,

$$w = \frac{1}{\|\mathbf{X}_S\|^2}\mathbf{X}_S^\top \mathbf{y}_S, \quad w' = \frac{1}{\|\mathbf{X}_{S'}\|^2}\mathbf{X}_{S'}^\top \mathbf{y}_{S'}$$

Then,

$$
\begin{aligned}
L(\mathbf{w}_S) &= \|\mathbf{y} - \mathbf{X}w\|^2 \\
&= \|\mathbf{y}_S - \mathbf{X}_S w\|^2 + \|\mathbf{y}_{S'} - \mathbf{X}_{S'}w\|^2 \\
&= \|\mathbf{y}_S\|^2 + w^2\|\mathbf{X}_S\|^2 - 2w\langle \mathbf{X}_S, \mathbf{y}_S\rangle + \|\mathbf{y}_{S'}\|^2 + w^2\|\mathbf{X}_{S'}\|^2 - 2w\langle \mathbf{X}_{S'}, \mathbf{y}_{S'}\rangle \\
&= \|\mathbf{y}_S\|^2 + \|\mathbf{y}_{S'}\|^2 + \frac{1}{\|\mathbf{X}_S\|^4}(\mathbf{X}_S^\top \mathbf{y}_S)^2\left(\|\mathbf{X}_S\|^2 + \|\mathbf{X}_{S'}\|^2\right) - \frac{2}{\|\mathbf{X}_S\|^2}\mathbf{X}_S^\top \mathbf{y}_S\left(\mathbf{X}_S^\top \mathbf{y}_S + \mathbf{X}_{S'}^\top \mathbf{y}_{S'}\right) \\
&= \|\mathbf{y}_S\|^2 + \|\mathbf{y}_{S'}\|^2 - \frac{(\mathbf{X}_S^\top \mathbf{y}_S)^2}{\|\mathbf{X}_S\|^2} + \frac{\|\mathbf{X}_{S'}\|^2(\mathbf{X}_S^\top \mathbf{y}_S)^2}{\|\mathbf{X}_S\|^4} - \frac{2\left(\mathbf{X}_S^\top \mathbf{y}_S\right)\left(\mathbf{X}_{S'}^\top \mathbf{y}_{S'}\right)}{\|\mathbf{X}_S\|^2} \\
&= \|\mathbf{y}_S\|^2 + \|\mathbf{y}_{S'}\|^2 + \frac{(\mathbf{X}_S^\top \mathbf{y}_S)^2}{\|\mathbf{X}_S\|^4}\left(\|\mathbf{X}_{S'}\|^2 - \|\mathbf{X}_S\|^2\right) - \frac{2\left(\mathbf{X}_S^\top \mathbf{y}_S\right)\left(\mathbf{X}_{S'}^\top \mathbf{y}_{S'}\right)}{\|\mathbf{X}_S\|^2}
\end{aligned}
$$

Similarly,

$$L(\mathbf{w}_{S'}) = \|\mathbf{y}_S\|^2 + \|\mathbf{y}_{S'}\|^2 + \frac{(\mathbf{X}_{S'}^\top \mathbf{y}_{S'})^2}{\|\mathbf{X}_{S'}\|^4}\left(\|\mathbf{X}_S\|^2 - \|\mathbf{X}_{S'}\|^2\right) - \frac{2\left(\mathbf{X}_{S'}^\top \mathbf{y}_{S'}\right)\left(\mathbf{X}_S^\top \mathbf{y}_S\right)}{\|\mathbf{X}_{S'}\|^2}$$

Subtracting,

$$
\begin{aligned}
L(\mathbf{w}_S) - L(\mathbf{w}_{S'}) &= \left[\frac{(\mathbf{X}_S^\top \mathbf{y}_S)^2}{\|\mathbf{X}_S\|^4} + \frac{(\mathbf{X}_{S'}^\top \mathbf{y}_{S'})^2}{\|\mathbf{X}_{S'}\|^4} - \frac{2(\mathbf{X}_S^\top \mathbf{y}_S)(\mathbf{X}_{S'}^\top \mathbf{y}_{S'})}{\|\mathbf{X}_S\|^2 \|\mathbf{X}_{S'}\|^2}\right]\left(\|\mathbf{X}_{S'}\|^2 - \|\mathbf{X}_S\|^2\right) \\
&= \left[\frac{\mathbf{X}_S^\top \mathbf{y}_S}{\|\mathbf{X}_S\|^2} - \frac{\mathbf{X}_{S'}^\top \mathbf{y}_{S'}}{\|\mathbf{X}_{S'}\|^2}\right]^2 \left(\mathrm{Vol}(\mathbf{X}_{S'})^2 - \mathrm{Vol}(\mathbf{X}_S)^2\right)
\end{aligned}
$$

The last step follows from the fact that $\mathrm{Vol}(\mathbf{X})^2 = |\mathbf{X}^\top\mathbf{X}| = \|\mathbf{X}\|^2$ when $d = 1$. Therefore, we have $L(\mathbf{w}_S) \leq L(\mathbf{w}_{S'})$ if and only if $\mathrm{Vol}(\mathbf{X}_S) \geq \mathrm{Vol}(\mathbf{X}_{S'})$. $\qquad\square$

***Alternate Proof of Proposition 3.*** Following Lemma 4, we denote $\Gamma := \mathbf{X}_S^\top \mathbf{y}_S$, $\Gamma' := \mathbf{X}_{S'}^\top \mathbf{y}_{S'}$ to consider $L(w) - L(w')$ as follows,

$$
\begin{aligned}
L(w) - L(w') &= [\|\mathbf{y}\|^2 - w^2(V_S^2 - V_{S'}^2) - 2w\Gamma'] - [\|\mathbf{y}\|^2 - w'^2(V_{S'}^2 - V_S^2) - 2w'\Gamma] \\
&= -w^2(V_S^2 - V_{S'}^2) + w'^2(V_{S'}^2 - V_S^2) - 2w\Gamma' + 2w'\Gamma \\
&= (w^2 + w'^2)(V_{S'}^2 - V_S^2) - 2(w\Gamma' - w'\Gamma) \\
&= (w^2 + w'^2)(V_{S'}^2 - V_S^2) - 2(w\frac{\Gamma'}{V_{S'}^2}V_{S'}^2 - w'\frac{\Gamma}{V_S^2}V_S^2) \\
&= (w^2 + w'^2)(V_{S'}^2 - V_S^2) - 2(ww'V_{S'}^2 - w'wV_S^2) \quad \text{Noting } w = \frac{\Gamma}{V_S^2}, w' = \frac{\Gamma'}{V_{S'}^2} \\
&= (w^2 + w'^2)(V_{S'}^2 - V_S^2) - 2ww'(V_{S'}^2 - V_S^2) \\
&= (w - w')^2(V_{S'}^2 - V_S^2)
\end{aligned}
$$

Since $(w - w')^2 \geq 0$, we have $L(w) - L(w') \geq 0 \iff V_{S'}^2 - V_S^2 \geq 0$ or equivalently, $L(w) \geq L(w') \iff V_{S'}^2 \geq V_S^2$. $\qquad\square$

**Lemma 4 (MSE of the least-squares solution on $\mathbf{X}_S$ for $d = 1$).** *Let $S, S'$ be a partition of the rows of the matrix $\mathbf{X}$, so that $\mathbf{X}_S, \mathbf{X}_{S'}$ are submatrices of $\mathbf{X}$, i.e. $\mathbf{X} = [\mathbf{X}_S^\top\ \mathbf{X}_{S'}^\top]^\top$. Let $\mathbf{y}_S, \mathbf{y}_{S'}$ be defined similarly. Further, let $w$ denote the least squares solution (note it is a scalar for $d = 1$) on the submatrix $\mathbf{X}_S$ with labels $\mathbf{y}_S$, i.e. $w = (\mathbf{X}_S^\top\mathbf{X}_S)^{-1}\mathbf{X}_S^\top\mathbf{y}_S = \Gamma/V_S^2$. Then the mean squared loss on the full matrix $L(w) := \|\mathbf{y} - \mathbf{X}w\|^2$ is*

$$L(w) = \|\mathbf{y}\|^2 - w^2(V_S^2 - V_{S'}^2) - 2w\Gamma' \tag{10}$$

*where $V_S := \mathrm{Vol}(\mathbf{X}_S), V_{S'} := \mathrm{Vol}(\mathbf{X}_{S'})$ and $\Gamma' := \mathbf{X}^\top\mathbf{y} - \mathbf{X}_S^\top\mathbf{y}_S$.*

***Proof of Lemma 4.*** For $d = 1$, observe the least squares solution $\mathbf{w} := (\mathbf{X}^\top \mathbf{X})^{-1} \mathbf{X}^\top \mathbf{y}$ is a scalar and $(\mathbf{X}^\top \mathbf{X})^{-1} = 1/(\|\mathbf{X}\|^2)$. Further, note that $\text{Vol}(\mathbf{X})^2 := \det(\mathbf{X}^\top \mathbf{X}) = \|\mathbf{X}\|^2$. Subsequently, the least squares solution $w$ for $(\mathbf{X}_S, \mathbf{y}_S)$ is $w = (\mathbf{X}_S^\top \mathbf{X}_S)^{-1} \mathbf{X}_S^\top \mathbf{y}_S = \Gamma/V_S^2$, where $\Gamma := \mathbf{X}_S^\top \mathbf{y}_S$, and let $w', \Gamma'$ be defined similarly for $(\mathbf{X}_{S'}, \mathbf{y}_{S'})$. Then we have

$$
\begin{aligned}
L(w) &= \|\mathbf{y} - \mathbf{X}w\|^2 \\
&= \|\mathbf{y}\|^2 - 2w\langle \mathbf{X}, \mathbf{y}\rangle + w^2 \|\mathbf{X}\|^2 \\
&= \|\mathbf{y}\|^2 - 2w\langle \mathbf{X}, \mathbf{y}\rangle + w^2 \|\mathbf{X}_S\|^2 + w^2 \|\mathbf{X}_{S'}\|^2 \\
&= \|\mathbf{y}\|^2 - 2w(\Gamma + \Gamma') + w^2 \|\mathbf{X}_S\|^2 + w^2 \|\mathbf{X}_{S'}\|^2 \\
&= \|\mathbf{y}\|^2 - 2w^2 V_S^2 - 2w\Gamma' + w^2 V_S^2 + w^2 V_{S'}^2 \\
&= \|\mathbf{y}\|^2 - w^2(V_S^2 - V_{S'}^2) - 2w\Gamma'.
\end{aligned}
$$

$\square$

***Derivation of*** (1). We will expand the $L(\mathbf{w}_S)$ into several terms and show there are common terms to both $L(\mathbf{w}_S)$ and $L(\mathbf{w}_{S'})$ which get canceled in $L(\mathbf{w}_S) - L(\mathbf{w}_{S'})$ and we analyze the difference in the remainder terms.

$$
L(\mathbf{w}_S) := \|\mathbf{y} - \mathbf{X}\mathbf{w}_S\|^2 = \|\mathbf{y}_S - \mathbf{X}_S\mathbf{w}_S\|^2 + \underbrace{\|\mathbf{y}_{S'} - \mathbf{X}_{S'}\mathbf{w}_S\|^2}_{A}
$$

Next, we want to write $A = \|\mathbf{y}_{S'} - \mathbf{X}_{S'}\mathbf{w}_{S'}\|^2 + R_S$, letting $R_S$ denote the remainder term. Note with this expression we have $L(\mathbf{w}_S) - L(\mathbf{w}_{S'}) = R_S - R_{S'}$. We first expand $A$ as follows,

$$
A = \underbrace{\|\mathbf{y}_{S'}\|^2}_{A_1} \underbrace{-2\langle \mathbf{y}_{S'}, \mathbf{X}_{S'}\mathbf{w}_S\rangle}_{B} + \underbrace{\|\mathbf{X}_{S'}\mathbf{w}_S\|^2}_{C}
$$

Next to rewrite $B, C$ as follows,

$$
B = -2\langle \mathbf{y}_{S'}, \mathbf{X}_{S'}(\mathbf{w}_{S'} - \mathbf{w}_{S'} + \mathbf{w}_S)\rangle = \underbrace{-2\langle \mathbf{y}_{S'}, \mathbf{X}_{S'}\mathbf{w}_{S'}\rangle}_{B_1} + \underbrace{-2\langle \mathbf{y}_{S'}, \mathbf{X}_{S'}(\mathbf{w}_S - \mathbf{w}_{S'})\rangle}_{B_2},
$$

$$
\begin{aligned}
C = \|\mathbf{X}_{S'}\mathbf{w}_S\|^2 &= \|\mathbf{X}_{S'}(\mathbf{w}_{S'} - \mathbf{w}_{S'} + \mathbf{w}_S)\|^2 \\
&= \underbrace{\|\mathbf{X}_{S'}\mathbf{w}_{S'}\|^2}_{C_1} + \underbrace{2\langle \mathbf{X}_{S'}\mathbf{w}_{S'}, \mathbf{X}_{S'}(\mathbf{w}_S - \mathbf{w}_{S'})\rangle + \|\mathbf{X}_{S'}(\mathbf{w}_S - \mathbf{w}_{S'})\|^2}_{C_2}.
\end{aligned}
$$

We can collect $A_1, B_1, C_1$ to complete the square of $\|\mathbf{y}_{S'} - \mathbf{X}_{S'}\mathbf{w}_{S'}\|^2$ and naturally collect $B_2, C_2$ to form the remainder $R_S$ as follows,

$$
\begin{aligned}
R_S &= -2\langle \mathbf{y}_{S'}, \mathbf{X}_{S'}(\mathbf{w}_S - \mathbf{w}_{S'})\rangle + 2\langle \mathbf{X}_{S'}\mathbf{w}_{S'}, \mathbf{X}_{S'}(\mathbf{w}_S - \mathbf{w}_{S'})\rangle + \|\mathbf{X}_{S'}(\mathbf{w}_S - \mathbf{w}_{S'})\|^2 \\
&= 2\langle \mathbf{X}_{S'}\mathbf{w}_{S'} - \mathbf{y}_{S'}, \mathbf{X}_{S'}(\mathbf{w}_S - \mathbf{w}_{S'})\rangle + \|\mathbf{X}_{S'}(\mathbf{w}_S - \mathbf{w}_{S'})\|^2 \\
&= \langle 2(\mathbf{X}_{S'}\mathbf{w}_{S'} - \mathbf{y}_{S'}) + \mathbf{X}_{S'}(\mathbf{w}_S - \mathbf{w}_{S'}), \mathbf{X}_{S'}(\mathbf{w}_S - \mathbf{w}_{S'})\rangle \\
&= \langle \mathbf{X}_{S'}(\mathbf{w}_S + \mathbf{w}_{S'}) - 2\mathbf{y}_{S'}, \mathbf{X}_{S'}(\mathbf{w}_S - \mathbf{w}_{S'})\rangle \\
&= \langle \mathbf{X}_{S'}^\top[\mathbf{X}_{S'}(\mathbf{w}_S + \mathbf{w}_{S'}) - 2\mathbf{y}_{S'}], \mathbf{w}_S - \mathbf{w}_{S'}\rangle.
\end{aligned}
$$

Now we can derive $R_S - R_{S'}$ as follows,

$$
\begin{aligned}
R_S - R_{S'} &= \langle \mathbf{X}_{S'}^\top[\mathbf{X}_{S'}(\mathbf{w}_S + \mathbf{w}_{S'}) - 2\mathbf{y}_{S'}], \mathbf{w}_S - \mathbf{w}_{S'}\rangle - \langle \mathbf{X}_S^\top[\mathbf{X}_S(\mathbf{w}_{S'} + \mathbf{w}_S) - 2\mathbf{y}_S], \mathbf{w}_{S'} - \mathbf{w}_S\rangle \\
&= \langle (\mathbf{X}_S^\top \mathbf{X}_S + \mathbf{X}_{S'}^\top \mathbf{X}_{S'})(\mathbf{w}_{S'} + \mathbf{w}_S) - 2\mathbf{X}_S^\top \mathbf{y}_S - 2\mathbf{X}_{S'}^\top \mathbf{y}_{S'}, \mathbf{w}_S - \mathbf{w}_{S'}\rangle \\
&= \langle (\mathbf{X}_S^\top \mathbf{X}_S + \mathbf{X}_{S'}^\top \mathbf{X}_{S'})(\mathbf{w}_{S'} + \mathbf{w}_S) - 2\mathbf{X}^\top \mathbf{y}, \mathbf{w}_S - \mathbf{w}_{S'}\rangle
\end{aligned}
$$

$\square$

**Adversarially constructed counter-examples to achieve arbitrary signs of $L(\mathbf{w}_S) - L(\mathbf{w}_{S'})$.** Given $\mathbf{X}_S, \mathbf{X}_{S'}$ as follows, we construct two sets of labels $\mathbf{y}_1, \mathbf{y}_2$ where $\mathbf{y}_1 = [\mathbf{y}_{1,S}^\top \ \mathbf{y}_{1,S'}^\top]^\top$ and $\mathbf{y}_2 = [\mathbf{y}_{2,S}^\top \ \mathbf{y}_{2,S'}^\top]^\top$ such that $L(\mathbf{w}_S) < L(\mathbf{w}_{S'})$ on $\mathbf{y}_1$ while $L(\mathbf{w}_S) > L(\mathbf{w}_{S'})$ on $\mathbf{y}_2$.

Note this example is adversarially constructed to show that a larger volume does not necessarily lead to a smaller MSE because $\text{Vol}()$ does *not* take the labels into consideration.

Let $d = 2$, $n = 6$ and fix $\mathbf{X}_S = \begin{bmatrix} 0.4197849, 0.82836752 \\ 0.8393158, 0.24545882 \\ 0.8544813, 0.72294841 \end{bmatrix}, \mathbf{X}_{S'} = \begin{bmatrix} 0.40205988, 0.44985846 \\ 0.36588236, 0.33433118 \\ 0.79521338, 0.34753677 \end{bmatrix}$.

Set two sets of labels $\mathbf{y}_1, \mathbf{y}_2$ as follows, $\mathbf{y}_{1,S} = \mathbf{0}$ and $\mathbf{y}_{1,S'} = \{\exp(10 \times x[1]) | x \in S'\}$; and $\mathbf{y}_{2,S} = \{\exp(10 \times x[1]) | x \in S\}$ and $\mathbf{y}_{2,S'} = \mathbf{0}$. The $\exp(10 \times x[1])$ refers to taking the exponential of the product between $10$ and the second value of a datum $x \in \mathbb{R}^2$. An observation is that since the true function/labels only depend on the second feature of each data point, the first feature is redundant/unnecessary to achieve a small MSE and yet included in the volume calculation. This is how we can construct the adversarial labels.

With this setting, we have $\text{Vol}(\mathbf{X}_S) > \text{Vol}(\mathbf{X}_{S'})$ and $L(\mathbf{w}_S) < L(\mathbf{w}_{S'})$ on $\mathbf{y}_1$ while $L(\mathbf{w}_S) > L(\mathbf{w}_{S'})$ on $\mathbf{y}_2$.

### A.3 Replication Robustness

***Proof of Lemma 1.*** Using the same notation for $\mathbf{X}_{\text{rep}} = [\mathbf{X}^\top \ \mathbf{x}_q^\top \ \dots \ \mathbf{x}_q^\top]^\top \in \mathbb{R}^{(n+m) \times d}$, we can write $\text{Vol}(\mathbf{X}_{\text{rep}})^2 := |\mathbf{X}_{\text{rep}}^\top \mathbf{X}_{\text{rep}}|$. Consider the simple case where $m = 1$, we have

$$\begin{aligned} |\mathbf{X}_{\text{rep}}^\top \mathbf{X}_{\text{rep}}| &= \left| [\mathbf{X}^\top \ \mathbf{x}_q^\top] \begin{bmatrix} \mathbf{X} \\ \mathbf{x}_q \end{bmatrix} \right| = |\mathbf{X}^\top \mathbf{X} + \mathbf{x}_q^\top \mathbf{x}_q| \\ &= (1 + \mathbf{x}_q (\mathbf{X}^\top \mathbf{X})^{-1} \mathbf{x}_q^\top) |\mathbf{X}^\top \mathbf{X}| \end{aligned}$$

where the last equality uses the matrix determinant lemma. Note with a full-rank $\mathbf{X}$, $\mathbf{X}^\top \mathbf{X}$ is invertible, symmetric and positive semi-definite, thus diagonalizable, as required by the matrix determinant lemma.

In general,

$$\begin{aligned} |\mathbf{X}_{\text{rep}}^\top \mathbf{X}_{\text{rep}}| &= \left| [\mathbf{X}^\top \ \underbrace{\mathbf{x}_q^\top \ \dots \ \mathbf{x}_q^\top}_{m \text{ terms}}] \begin{bmatrix} \mathbf{X} \\ \mathbf{x}_q \\ \vdots \\ \mathbf{x}_q \end{bmatrix} \right| = |\mathbf{X}^\top \mathbf{X} + m \times \mathbf{x}_q^\top \mathbf{x}_q| \\ &= (1 + m \times \mathbf{x}_q (\mathbf{X}^\top \mathbf{X})^{-1} \mathbf{x}_q^\top) |\mathbf{X}^\top \mathbf{X}| \end{aligned}$$

In other words, $\text{Vol}(\mathbf{X}_{\text{rep}}) = \text{Vol}(\mathbf{X}) \times (1 + m \times \mathbf{x}_q (\mathbf{X}^\top \mathbf{X})^{-1} \mathbf{x}_q^\top)^{1/2}$. $\qquad\square$

**Lemma 5** (**Inflation vs. $\alpha$**). *When $\alpha = 1/(\beta n)$, the inflation of replicating $\mathbf{X} \in \mathbb{R}^{n \times d}$ with a replication factor $c$ is upper bounded as follows,*

$$\lim_{n \to \infty} \left( \sum_{p=0}^{nc} (\frac{1}{\beta n})^p \right)^n = \exp(\beta^{-1}).$$

***Proof of Lemma 5.*** Suppose some rows of $\mathbf{X}$ are copied and appended back to get $\mathbf{X}_{\text{rep}} \in \mathbb{R}^{(nc) \times d}$ such that the replication factor is $c$. For simplicity, we set $\omega$ such that each $d$-cube contains identical data points. As there are $n$ rows in the original $\mathbf{X}$, there can be at most $n$ non-empty $d$-cubes for any $\omega$ by the pigeon-hole principle. Note since the replication is by direct copying, the number of non-empty $d$-cubes for $\mathbf{X}_{\text{rep}}$ is upper bounded by $n$.

We next consider the inflation, which is the ratio $\mathrm{RV}(\mathbf{X}_{\mathrm{rep}};\omega)/\mathrm{RV}(\mathbf{X};\omega)$ as follows,

$$
\begin{aligned}
\frac{\mathrm{RV}(\mathbf{X}_{\mathrm{rep}};\omega)}{\mathrm{RV}(\mathbf{X};\omega)} &= \frac{\mathrm{Vol}(\widetilde{\mathbf{X}}_{\mathrm{rep}}) \times \prod_{i\in\Omega_{\mathrm{rep}}} \rho_{i,\mathrm{rep}}}{\mathrm{Vol}(\widetilde{\mathbf{X}}) \times \prod_{i\in\Omega} \rho_i} \\
&= \frac{\prod_{i\in\Omega_{\mathrm{rep}}} \rho_{i,\mathrm{rep}}}{\prod_{i\in\Omega} \rho_i} \\
&\leq \frac{\prod_{i\in\Omega_{\mathrm{rep}}} \rho_{i,\mathrm{rep}}}{1} \triangleq \prod_{i\in\Omega_{\mathrm{rep}}} \sum_{p=0}^{\phi_{i,\mathrm{rep}}} \alpha^p \\
&\leq \prod_{i\in\Omega_{\mathrm{rep}}} \sum_{p=0}^{nc} \alpha^p \triangleq \prod_{i\in\Omega_{\mathrm{rep}}} \sum_{p=0}^{nc} (\frac{1}{\beta n})^p \\
&\leq \left( \sum_{p=0}^{nc} (\frac{1}{\beta n})^p \right)^n
\end{aligned}
$$

The first line is by definition; the second equality is by observing that $\widetilde{\mathbf{X}}_{\mathrm{rep}} = \widetilde{\mathbf{X}}$ due to direct copying and that each $d$-cube contains identical data points; the next inequality is by observing $\prod_{i\in\Omega} \rho_i \geq 1$; the next equality is by definition of $\rho_i$; the next inequality uses $nc$ to upper bound the number of data points in any $d$-cube; the next equality substitutes $\alpha = 1/(\beta n)$ and the last inequality is by bounding the number of non-empty $d$-cubes by $n$.

We upper-bound $\sum_{p=0}^{nc}(1/\beta n)^p$ with respect to $c \to \infty$ as follows,

$$
\sum_{p=0}^{nc} (\frac{1}{\beta n})^p \leq \frac{1}{1 - \frac{1}{\beta n}} = \frac{\beta n}{\beta n - 1} = 1 + \frac{1}{\beta n - 1}.
$$

Next, apply the limit of $n \to \infty$ to give the following [2]:

$$
\lim_{n\to\infty} (1 + \frac{1}{\beta n - 1})^n = \lim_{n\to\infty} (1 + \frac{1}{n} \times \frac{1}{\beta - \frac{1}{n}})^n = \exp(\beta^{-1}).
$$

The last equality is by first considering $(\beta - \frac{1}{n})^{-1} \to \beta^{-1}$ as $n \to \infty$ and then using a known result of $\lim_{n\to\infty} (1 + \frac{x}{n})^n = \exp(x) \; \forall x$. $\qquad \square$

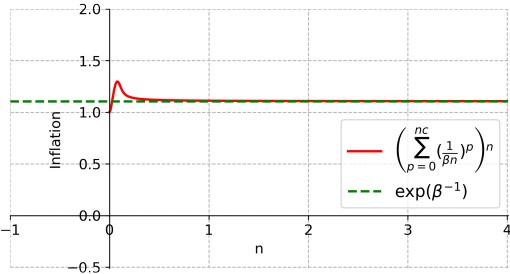

Figure 9: Inflation vs. $n$. With $\beta = 10, c = 100$, the inflation (red) quickly decays and converges to below the green line where $\exp(\beta^{-1}) \approx 1.105$. Note $x$-axis denotes $n$, the number of data points.

Note Lemma 5 upper bounds the inflation is with respect to the replication factor $c$, while Lemma 6 upper bounds the RV of a $\mathbf{X}$ which may not contain replicated data.

**Lemma 6** ($\mathrm{RV}(\cdot;\omega)$ **vs.** $\omega$). *The growth of $\mathrm{RV}(\mathbf{X};\omega)$ with respect to $\omega$ is slow and upper bounded as follows,*

$$
\sup_{\omega} \lim_{n\to\infty} \frac{\mathrm{RV}(\mathbf{X};\omega)}{\mathrm{Vol}(\mathbf{X})} \leq \exp(\beta^{-1})
$$

---

[2]The asymptotic condition on $n$ is included for theoretical rigor and may be easily removed in practice for any reasonable $n$ (larger than 10). See Fig. 9.

where $n$ is the number of rows in $\mathbf{X}$ and $\beta = (\alpha n)^{-1}$ where $\alpha$ is a user-determined robustness coefficient.

***Proof of Lemma 6.*** Recall $\mathrm{RV}(\mathbf{X};\omega) := \mathrm{Vol}(\widetilde{\mathbf{X}}) \times \prod_{i\in\Omega} \rho_i$. We assume $\mathrm{Vol}(\widetilde{\mathbf{X}}) \leq \mathrm{Vol}(\mathbf{X})$ because $\widetilde{\mathbf{X}}$ possibly has a smaller number of rows. Due to Lemma 1, we know that reducing a row decreases the total volume. We subsequently verify the result of Lemma 6 to confirm this assumption is satisfied.

To derive an upper bound on $\prod_{i\in\Omega} \rho_i$, we first consider the fact that given an $\omega$, the number of non-empty $d$-cubes is upper bounded by $n$, the number of rows in $\mathbf{X}$. Then, for each of these $d$-cubes, the up-weight constant $\rho_i$ is upper bounded by $1/(1-\alpha)$ due to the geometric series sum. Therefore,

$$\prod_{i\in\Omega} \rho_i \leq (\frac{1}{1 - \frac{1}{\beta n}})^n.$$

Using the similar technique in proof of Lemma 5, we have

$$\lim_{n\to\infty} (\frac{1}{1 - \frac{1}{\beta n}})^n = \exp(\beta^{-1}),$$

and the result follows.

Fig. 10 shows a numerical experiment where 50 matrices $\mathbf{X}$ are independently and randomly drawn, and for each we compute the ratio $\mathrm{RV}(\mathbf{X};\omega)/\mathrm{Vol}(\mathbf{X})$ (in $y$-axis) over a range of $\omega$ (in $x$-axis). In particular, $\beta = 10$ and we see the upper bound is in fact followed. $\square$

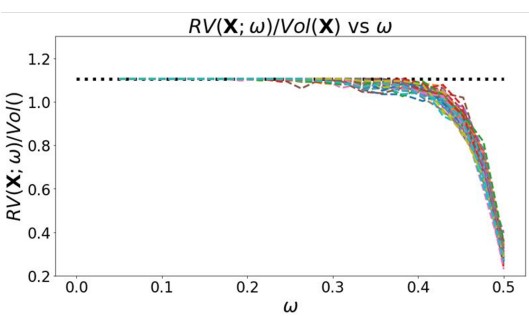

Figure 10: Growth of $\mathrm{RV}(\mathbf{X};\omega)$ is upper bounded by $\exp(\beta^{-1})$ (black dotted line). $\beta = 10$ and the discretization coefficient $\omega$ takes range $(0, 0.5)$.

***Proof of Proposition 5 (Bounded Distortion).*** We re-arrange the distortion $\delta(\omega) := [\mathrm{RV}(\mathbf{X}_S;\omega)/\mathrm{RV}(\mathbf{X}_{S'};\omega)]/[\mathrm{Vol}(\mathbf{X}_S)/\mathrm{Vol}(\mathbf{X}_{S'})]$ as follows,

$$\delta(\omega) = \underbrace{\frac{\mathrm{RV}(\mathbf{X}_S;\omega)}{\mathrm{Vol}(\mathbf{X}_S)}}_{\leq \exp(\beta^{-1}) \text{ by Lemma 6}} \times \underbrace{\frac{\mathrm{Vol}(\mathbf{X}_{S'})}{\mathrm{RV}(\mathbf{X}_{S'};\omega)}}_{\leq 1 \text{ by Proposition 8}} \leq \exp(\beta^{-1}). \tag{11}$$

The $\leq 1$ is from the fact that replication-robustness valuation on an original matrix (without replication) is no smaller than the original volume (with two mild conditions that $\mathbf{X}$ contains sufficient diversity to *not* resemble a replicated dataset, and $\alpha$ is not too small, see Proposition 8, we empirically verify these assumptions by verifying the upper and lower bounds of $\delta(\omega)$ in Fig. 11), implying the RV will not reduce the information contained in the original matrix.

The lower bound of $(\exp(\beta^{-1}))^{-1}$ is by an argument by symmetry. Specifically, taking reciprocal on both sides of

$$\frac{\mathrm{RV}(\mathbf{X}_S;\omega)}{\mathrm{Vol}(\mathbf{X}_S)} \times \frac{\mathrm{Vol}(\mathbf{X}_{S'})}{\mathrm{RV}(\mathbf{X}_{S'};\omega)} \leq \exp(\beta^{-1})$$

gives

$$\frac{\mathrm{RV}(\mathbf{X}_{S'};\omega)}{\mathrm{Vol}(\mathbf{X}_{S'})} \times \frac{\mathrm{Vol}(\mathbf{X}_S)}{\mathrm{RV}(\mathbf{X}_S;\omega)} \geq (\exp(\beta^{-1}))^{-1}.$$

Since the reference on $S, S'$ is arbitrary, we arrive at the lower bound by switching the indexing.

Furthermore, we verify the two conditions required to apply Proposition 8 by empirically verifying the distortion is bounded. We construct two equal-sized, independent and identically sampled $d$-dimensional matrices, $\mathbf{X}_S, \mathbf{X}_{S'}$ and plot the distortion over a range of $\omega$. Note we have considered varied $d \in \{1, 2, 5, 10\}$ and two such distributions: $d$ dimensional $\mathcal{N}(0, 1)$ or uniform distribution $U(0, 1)$. The result is in Fig. 11. The black dotted lines are the theoretical upper and lower bounds. Fig. 11 suggests the bound on distortion may be tighter, implying in practice the consistency in relative valuation is preserved. $\square$

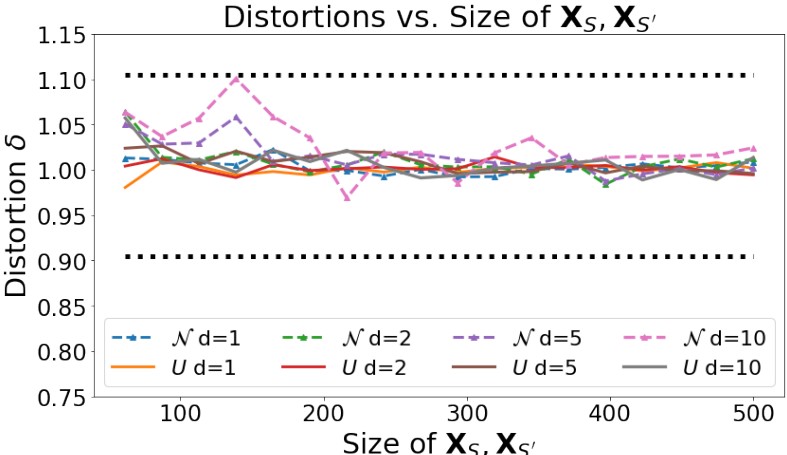

Figure 11: Distortion $\delta(\omega)$ vs. size of $\mathbf{X}_S, \mathbf{X}_{S'}$. $\mathbf{X}_S, \mathbf{X}_{S'}$ are equal-sized, independent and identically sampled from $d$-dimensional normal distribution $\mathcal{N}(0, 1)$ or uniform distribution $U(0, 1)$. Black dotted lines are $\exp(\beta^{-1})^{-1}, \exp(\beta^{-1})$ where $\beta = 10$. The discretization coefficient $\omega = 0.1$.

**Proof of Proposition 6.** By definition and simple rearranging, we can write

$$\gamma_{\mathrm{RV}} \triangleq \frac{\mathrm{RV}(\mathbf{X}; \omega)}{\sup_c \mathrm{RV}(\mathrm{replicate}(\mathbf{X}, c); \omega)} = \frac{\mathrm{Vol}(\widetilde{\mathbf{X}})}{\mathrm{Vol}(\widetilde{\mathbf{X}}_{\mathrm{rep}})} \times \prod_{i \in \Psi} \frac{\rho_i}{\rho_i'}$$

where $\rho_i'$ denotes the coefficient for the $d$-cube after replication. We calculate these two terms separately.

Consider $\boldsymbol{\mu}_i$ for some $d$-cube, and let $\boldsymbol{\mu}_i'$ denote the statistic after replication. We have $\boldsymbol{\mu}_i = \boldsymbol{\mu}_i'$ because each data point in the $d$-cube is replicated for equal number of times (i.e., $\to \infty$) due to $\sup_c$. This implies $\widetilde{\mathbf{X}} = \widetilde{\mathbf{X}}_{\mathrm{rep}}$ and $\mathrm{Vol}(\widetilde{\mathbf{X}}) / \mathrm{Vol}(\widetilde{\mathbf{X}}_{\mathrm{rep}}) = 1$.

For a non-empty $d$-cube, $\rho_i \geq 1$ and $\rho_i' \leq 1/(1 - \alpha)$ due to the geometric series. So we have $\frac{\rho_i}{\rho_i'} \geq (1 - \alpha)$. Multiplying all $|\Psi|$ such ratios we have $\prod_{i \in \Psi} \frac{\rho_i}{\rho_i'} \geq (1 - \alpha)^{|\Psi|}$. Combining this with the previous result completes the proof. $\square$

**Proof of Proposition 7.** We outline the proof ideas and omit the tedious details.

1. Reduction of $\mathrm{RV}(\cdot; \omega)$ to $\mathrm{Vol}()$ requires careful tracing of the constants $K_{\widetilde{\mathbf{X}}, i}$ using Lemma 1. There may be different ways to set the values of $K_{\widetilde{\mathbf{X}}, i}$ for this equality to hold from different order of tracing. We provide one such construction.

   First, construct $\mathbf{A} = \widetilde{\mathbf{X}}$. We will use $\mathbf{A}$ to represent the intermediate construction of $\mathbf{X}$ for the calculation of $\mathrm{Vol}()$. Also, set $K_{\widetilde{\mathbf{X}}, i} = 1, i \in \Psi$. This step fills up each non-empty $d$-cube with one data point.

   The second step exhausts the remaining data points $\mathbf{B}$, one distinct data point at a time. The remaining data points from the first step are all the data points excluding one data

point for each $d$-cube. We abuse notation a little to write $\mathbf{B} := \mathbf{X} \setminus \mathbf{A} = \mathbf{X} \setminus \widetilde{\mathbf{X}}$. For each distinct $\mathbf{x} \in \mathbf{B}$, suppose it goes into the $i$-th $d$-cube and it has $m$ copies in $\mathbf{B}$: update $K_{\widetilde{\mathbf{X}},i} \leftarrow (1 + m \times \mathbf{x}(\mathbf{A}^\top \mathbf{A})^{-1}\mathbf{x}^\top)^{1/2}$ and then update $\mathbf{A} \leftarrow [\mathbf{A}^\top \underbrace{\mathbf{x}^\top \dots \mathbf{x}^\top}_{m \text{ times}}]^\top$ by Lemma 1

and remove all copies of $\mathbf{x}$ from $\mathbf{B}$. Iterate until $\mathbf{B}$ is empty.

2. $\gamma_{\mathrm{RV}_1} = 1$ is from a direct application of Proposition 6.

$\square$

**Proposition 8** (**Conditions for** $\mathrm{RV}(\mathbf{X}; \omega) \geq \mathrm{Vol}(\mathbf{X})$). *For a given $\omega$ and $\mathbf{X}$, $\mathrm{RV}(\mathbf{X}; \omega) \geq \mathrm{Vol}(\mathbf{X})$ requires the following two conditions:*

1. *The original $\mathbf{X}$ contains sufficient diversity that it does not resemble replicated copies of data.*

2. *The $\alpha$ is not too small (so $\prod_{\omega_i} \rho_i$ will not be too small).*

***Proof of Proposition 8***. For a given $\omega$, let the rows which occupy $d$-cubes alone be in $\mathbf{X}_A$ and the rest in $\mathbf{X}_B$. Each row in $\mathbf{X}_A$ occupies some $d$-cube by itself, while each row in $\mathbf{X}_B$ has to "share" a $d$-cube with another row in $\mathbf{X}_B$. With this, we rearrange and rewrite $\mathbf{X} = \begin{bmatrix} \mathbf{X}_A \\ \mathbf{X}_B \end{bmatrix}$ and we have

$$\mathrm{Vol}(\mathbf{X})^2 = |\mathbf{G}_A| \times |\mathbf{I} + \mathbf{G}_A^{-1}\mathbf{G}_B| \tag{12}$$

from Lemma 7 where $\mathbf{G}_A := \mathbf{X}_A^\top \mathbf{X}_A, \mathbf{G}_B := \mathbf{X}_B^\top \mathbf{X}_B$.

Let $\widetilde{\mathbf{X}}_B$ be the estimated $\mathbf{X}_B$ according to the $d$-cubes: for each $d$-cube containing more than one row, take the average of the rows in a $d$-cube, and put this as one row in $\widetilde{\mathbf{X}}_B$. With this, we write $\widetilde{\mathbf{X}} = \begin{bmatrix} \mathbf{X}_A \\ \widetilde{\mathbf{X}}_B \end{bmatrix}$. Note $\widetilde{\mathbf{X}}$ is part of the calculation for $\mathrm{RV}(\mathbf{X}; \omega)$.

We have

$$\mathrm{Vol}(\widetilde{\mathbf{X}})^2 = |\mathbf{G}_A| \times |\mathbf{I} + \mathbf{G}_A^{-1}\widetilde{\mathbf{G}}_B|$$

similarly as above where $\widetilde{\mathbf{G}}_B := \widetilde{\mathbf{X}}_B^\top \widetilde{\mathbf{X}}_B$. Substituting this into the definition of RV gives

$$\mathrm{RV}(\mathbf{X}) = \mathrm{Vol}(\widetilde{\mathbf{X}}) \times (\prod_{i \in \Omega} \rho_i) = |\mathbf{G}_A| \times |\mathbf{I} + \mathbf{G}_A^{-1}\widetilde{\mathbf{G}}_B| \times (1 + \epsilon) \tag{13}$$

Here, we write $\prod_{i \in \Omega} \rho_i = (1 + \epsilon)$ where $\epsilon \geq 0$ is a constant which depends on $\alpha$. A smaller $\alpha$ leads to a smaller $\epsilon$.

We compare (12) and (13) as follows,

$$|\mathbf{G}_A| \times |\mathbf{I} + \mathbf{G}_A^{-1}\mathbf{G}_B| \leq |\mathbf{G}_A| \times |\mathbf{I} + \mathbf{G}_A^{-1}\widetilde{\mathbf{G}}_B| \times (1 + \epsilon)$$
$$|\mathbf{I} + \mathbf{G}_A^{-1}\mathbf{G}_B| \leq |\mathbf{I} + \mathbf{G}_A^{-1}\widetilde{\mathbf{G}}_B| \times (1 + \epsilon).$$

While we want to find the conditions under which above inequality is satisfied, it may be more intuitive to consider the conditions which can dissatisfy it. So we consider

$$|\mathbf{I} + \mathbf{G}_A^{-1}\mathbf{G}_B| > |\mathbf{I} + \mathbf{G}_A^{-1}\mathbf{G}_{\hat{B}}| \times (1 + \epsilon).$$

Due to the previous discussion in Lemma 7, we know $|\mathbf{I} + \mathbf{G}_A^{-1}\mathbf{G}_B|$ is large if $\mathbf{X}_A$ covers little relative to $\mathbf{X}_B$, similarly for $|\mathbf{I} + \mathbf{G}_A^{-1}\widetilde{\mathbf{G}}_B|$. Therefore, we are left to find out the conditions where $\mathbf{X}_A$ covers little relative to $\mathbf{X}_B$ but $\mathbf{X}_A$ covers a lot relative to $\widetilde{\mathbf{X}}_B$ as this will lead to the l.h.s to be larger than r.h.s. The extreme case of $\mathbf{X}_A$ is empty while $\mathbf{X}_B$ contains multiple copies of the same datum, fits the described scenario very well.

In summary, under two conditions: 1) the original $\mathbf{X}$ contains sufficient diversity that it does not resemble replicated copies of data; 2) the $\alpha$ is not too small (so $\prod_{\omega_i} \rho_i$ will not be too small); then we have $\mathrm{RV}(\mathbf{X}; \omega) \geq \mathrm{Vol}(\mathbf{X})$. The first condition is intuitive. The second one on $\alpha$ can be understood

as a trade-off between enforcing robustness (smaller $\alpha$) vs. representing data (larger $\alpha$). In practice, we should set $\alpha$ to achieve the desired robustness guarantee but not extremely small as it would have an over-correcting effect of mistakenly reducing the value of an honest dataset.

$\square$

# B   Additional Experimental Results

## B.1   Dataset License

Credit Card [2]: Database Contents License (DbCL); Uber & Lyft [5]: CC0 1.0 Universal (CC0 1.0); Used Car [1]: CC0 1.0 Universal (CC0 1.0); Hotel Reviews [4]: Attribution-NonCommercial 4.0 International (CC BY-NC 4.0); CaliH [20]: CC0 1.0 Universal (CC0 1.0); KingH [3]: CC0 1.0 Universal (CC0 1.0); USCensus [6]: CC0 1.0 Universal (CC0 1.0); The FaceA dataset [41]: non-commercial research purposes only.

## B.2   Simulation for Replication Experiments on Volume and RV

We illustrate that the inflation of robust volumes (RV) can be controlled. We generate a dataset with 200 i.i.d. samples uniformly drawn from the synthetic 6D Friedman [13] function. As shown in Fig. 12, the volume of the dataset explodes exponentially with the number of full dataset replications. On the contrary, robust volume controls the volume explosion through $\alpha$ and the resultant RV stays almost constant with replications. Similar behavior is observed when we randomly select data rows to replicate instead.

In a more realistic adversarial replication setting, random noises could be injected into the replicated data rows. Interestingly, RV remains robust when the magnitude of the injected noise is small relative to $\omega$, further demonstrating RV's practical utility. Specifically in this experimental setting when $\omega$ is set to $0.1$, the RV is kept almost constant when the random Gaussian noise has $\sigma = 0.01$ (see pink dashed line in Fig. 12). On the other hand, RV does not inflate as much as Vol when $\sigma$ is increased to $0.03$ (compare gray and red lines). However, we observe if $\sigma$ becomes too large, the robustness degrades. This is because when the magnitude of injected noise is larger than the actual data, it effectively becomes "new" but noisy data. Consequently, it is still practically challenging when the Gaussian noise is large. As large noise could dilute the data row's original information, it is difficult to distinguish whether the row is replicated (with noise).

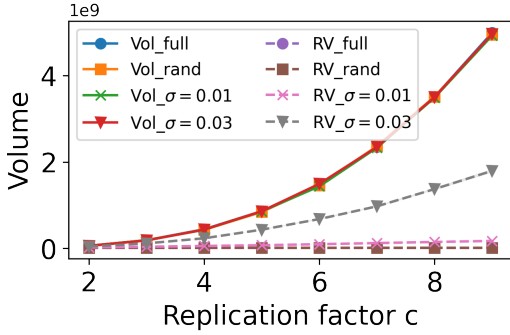

Figure 12: Vol & RV vs. replication. `full`, `rand` and $\sigma$ denote three types of replications, where `full` (*resp.* `rand`) replicates all data (*resp.* random rows) and $\sigma$ denotes noisy replication with noise $\sim \mathcal{N}(0, \sigma^2)$. Here, $\omega = 0.1$.

## B.3   Selection of the Discretization Coefficient

Throughout the work, we set $\omega = 0.1$ for standardized features, including for real-world datasets which may contain unknown noise in labels. Through synthetic experiments, we show empirically that $\omega \in [0, 0.5]$ is a suitable range for standardized features when noise is small. Intuitively, feature standardization "squashes" most of the data to a relatively small range. For instance, the $[-2, 2]$

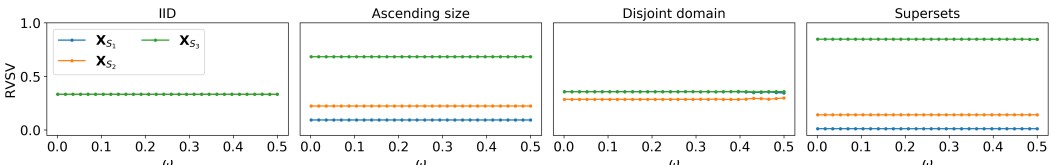

Figure 13: The choice of $\omega$ does not change values much under (A) i.i.d., (B) ascending dataset size, (C) disjoint input domain and (D) supersets. Setting $\omega > 0.5$ is not recommended as it results in an overly "compressed" $\widetilde{\mathbf{X}}$.

range is the $95\%$ confidence interval for standardized features. As illustrated in Fig. 13, the relative RVSV does not vary much for any $\omega < 0.5$ under all 4 settings: (A) i.i.d., (B) ascending dataset sizes, (C) disjoint input domain and (D) supersets.

Note that in the disjoint input domain setting, $\mathbf{X}_{S_1}$ and $\mathbf{X}_{S_3}$ are valued more because of feature standardization. The standardized features have the following interpretation: values that are very positive or negative are statistically rare, and thus are more valuable. In contrast, values which are close to the mean (a value of 0) are statistically common, and thus less valuable. The implication is less common data are given higher values. We find $\omega = 0.1$ suitable under standard scaling and in practice it may be adjusted based on the prior on the amount of noise expected in the feature/input, i.e., larger noise requires a larger $\omega$.

## B.4 Robust Volume and Learning Performance

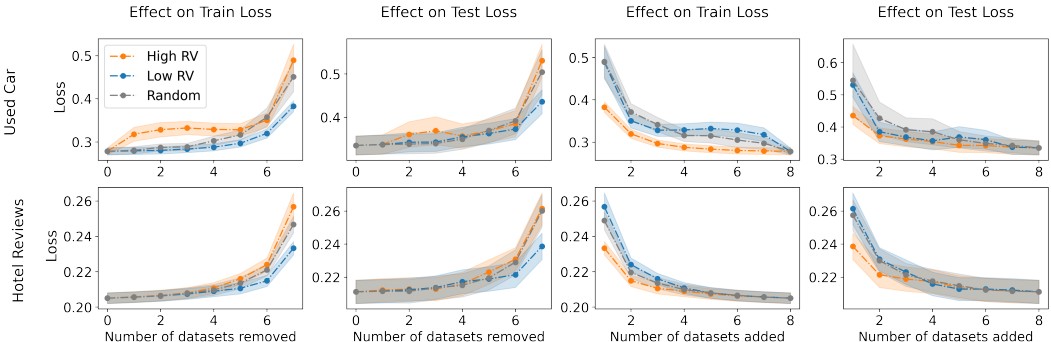

Figure 14: The effect of removing/adding the dataset with the highest/lowest RV on the train/test loss for two additional real-world datasets, the UK used car and the hotel reviews. The plots show the average and standard errors over 50 random trials.

To further verify that a larger RV leads to a better learning performance, we present results on two additional real-world datasets, the UK used car dataset [1] (i.e., car price prediction) and the Trip Advisor hotel reviews dataset [4] (i.e., numerical rating prediction). The two datasets are pre-processed through feature selection or neural networks to contain 5 and 8 standardized features respectively. Other experimental setups follow that of Sec. 5.1. The results are in Fig. 14. We observe a consistent general trend, adding (*resp.* removing) a dataset with high RV leads to lower (*resp.* higher) train and test loss. These results (along with the previous results) suggest that RV is a good indicator for the learning performance on the data without requiring validation, and is thus a good data valuation method.

## B.5 Overlap of Input Domains

To extend the "disjoint input domain" case discussed in Sec. 5.2, we set the input range of $\mathbf{X}_{S_1}, \mathbf{X}_{S_2}$ to be $[0, 0.5 + z]^6$ and $\mathbf{X}_{S_3}$ to be $[0.5, 1]^6$, where $z$ is the amount of domain overlap. To interpret, a larger $z$ implies a less "unique" dataset $\mathbf{X}_{S_3}$ and thus less value for $\mathbf{X}_{S_3}$. We show in Fig. 15 that all methods including RVSV observe the correct trend, except that VLSV is still dominated by $\nu(\emptyset)$

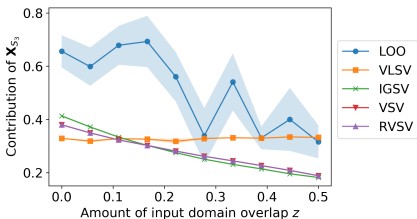

Figure 15: The effect of an increasing amount of input domain overlap on the valuation of $\mathbf{X}_{S_3}$, across different valuation methods. The values are averaged over 10 trails.

discussed in Sec. 5.2. Interestingly, we also observe a similar rate of decrease of the relative value of $\mathbf{X}_{S_3}$ for IGSV, VSV and RVSV.

### B.6 Replication Experiments

**Experimental Settings**. For CaliH and KingH, we train a neural network (NN) with 2 hidden layers consisting of 64 and 10 hidden units, respectively. For USCensus, we train an NN with 2 hidden layers consisting of 128 and 16 hidden units, respectively. For FaceA, we train a convolutional NN with 3 convolutional layers (the first two of which each followed by 2-dimensional batch normalization and max pooling), followed by 3 fully connected layers consisting of 1024, 64, and 10 hidden units respectively. The activation function used is the rectified linear unit (ReLU). Consequently, the features after the last hidden layer may contain completely zeros, so we remove features that contain only zero. Alternatively, leaky ReLU can be used instead to prevent this issue.

**Additional Results**. The results on four datasets (four rows) CaliH, KingH, USCensus and FaceA for the two non-i.i.d. distributions: *supersets* and *disjoint* are in Fig. 16. We can make these observations from the results: LOO's behavior is unstable, in particular for supersets ratio of 0.1 (leftmost column) where 10% of $\mathbf{X}_{S_1}$ is contained in $\mathbf{X}_{S_3}$. VSV and IGSV both increase quite quickly for $c \leq 20$. RVSV and VLSV are consistent regardless of $c$. The tabulated SV similarity results for KingH, USCensus and FaceA are in Tables 2, 3 and 4. We find that RVSV generally performs the best, in terms of similarity in relative valuations to the validation-based VLSV.

Table 2: Similarity with VLSV under replication for **KingH**. Bold values indicate best results.

| Method | i.i.d. | | | disjoint 0 | | | disjoint 1 | | | supersets 0.1 | | | supersets 1 | | |
|---|---|---|---|---|---|---|---|---|---|---|---|---|---|---|---|
| | $r_p$ | cos | $1/l_2$ | $r_p$ | cos | $1/l_2$ | $r_p$ | cos | $1/l_2$ | $r_p$ | cos | $1/l_2$ | $r_p$ | cos | $1/l_2$ |
| LOO | 0.693 | 0.215 | 0.381 | -0.008 | 0.483 | 0.964 | **0.704** | 0.752 | 1.971 | -0.114 | 0.051 | 0.091 | **0.883** | 0.524 | 1.068 |
| IGSV | -0.898 | 0.639 | 1.569 | -0.937 | 0.637 | 1.547 | -0.998 | 0.629 | 1.520 | -0.962 | 0.641 | 1.597 | 0.592 | 0.660 | 1.723 |
| VSV | -0.902 | 0.741 | 2.055 | -0.932 | 0.743 | 2.084 | -0.999 | 0.727 | 1.954 | -0.963 | 0.750 | 2.140 | 0.547 | 0.782 | 2.415 |
| RVSV-005 | **0.819** | **0.985** | **9.890** | 0.283 | **0.998** | **28.557** | -0.908 | **0.998** | **26.040** | 0.668 | **0.980** | **8.425** | -0.978 | **0.940** | **4.757** |
| RVSV-01 | 0.817 | 0.977 | 7.965 | -0.000 | 0.996 | 19.873 | -0.874 | 0.995 | 16.779 | 0.662 | 0.962 | 6.109 | -0.977 | 0.892 | 3.416 |

Table 3: Similarity with VLSV under replication for **USCensus**. Bold values indicate best results. The disjoint ratio 0 for USCensus dataset leads to $NaN$ values for VSV so we show from disjoint ratio 0.2.

| Method | i.i.d. | | | disjoint 0.2 | | | disjoint 1 | | | supersets 0.1 | | | supersets 1 | | |
|---|---|---|---|---|---|---|---|---|---|---|---|---|---|---|---|
| | $r_p$ | cos | $1/l_2$ | $r_p$ | cos | $1/l_2$ | $r_p$ | cos | $1/l_2$ | $r_p$ | cos | $1/l_2$ | $r_p$ | cos | $1/l_2$ |
| LOO | **0.891** | 0.816 | 2.493 | -0.127 | 0.517 | 1.632 | -0.950 | 0.164 | 0.290 | **0.845** | 0.211 | 0.396 | **0.682** | 0.602 | 1.344 |
| IGSV | 0.302 | 0.640 | 1.582 | 0.589 | 0.640 | 1.579 | -0.999 | 0.641 | 1.599 | 0.475 | 0.640 | 1.596 | -0.022 | 0.658 | 1.713 |
| VSV | 0.292 | 0.739 | 2.064 | 0.606 | 0.739 | 2.052 | -0.998 | 0.739 | 2.073 | 0.482 | 0.740 | 2.061 | -0.099 | 0.779 | 2.368 |
| RVSV-005 | -0.975 | **0.963** | **6.223** | 0.758 | **0.998** | **30.976** | 0.687 | **0.999** | **46.907** | -0.999 | **0.974** | **7.456** | -0.846 | **0.915** | **3.920** |
| RVSV-01 | -0.976 | 0.958 | 5.782 | 0.756 | 0.997 | 24.403 | 0.602 | 0.998 | 28.317 | -0.999 | 0.973 | 7.257 | -0.847 | 0.908 | 3.753 |

Table 4: Similarity with VLSV under replication for **FaceA**. Bold values indicate best results.

| Method | i.i.d. | | | disjoint 0 | | | disjoint 1 | | | supersets 0.1 | | | supersets 1 | | |
|---|---|---|---|---|---|---|---|---|---|---|---|---|---|---|---|
| | $r_p$ | cos | $1/l_2$ | $r_p$ | cos | $1/l_2$ | $r_p$ | cos | $1/l_2$ | $r_p$ | cos | $1/l_2$ | $r_p$ | cos | $1/l_2$ |
| LOO | -0.080 | 0.558 | 1.165 | **0.823** | 0.709 | 1.854 | 0.903 | 0.690 | 1.662 | -1.000 | 0.948 | 5.157 | **0.693** | **0.946** | **5.084** |
| IGSV | 0.019 | 0.732 | 2.088 | 0.229 | 0.733 | 2.100 | -0.898 | 0.731 | 2.096 | -0.838 | 0.735 | 2.137 | 0.632 | 0.760 | 2.347 |
| VSV | 0.013 | 0.701 | 1.807 | 0.226 | 0.706 | 1.847 | -0.894 | 0.702 | 1.816 | -0.836 | 0.709 | 1.869 | 0.603 | 0.745 | 2.095 |
| RVSV-005 | -0.867 | **0.941** | **4.828** | 0.340 | **1.000** | **69.502** | 0.923 | **1.000** | **100.947** | 0.869 | **0.943** | **4.909** | -0.957 | 0.880 | 3.209 |
| RVSV-01 | -0.877 | 0.884 | 3.275 | 0.140 | 0.997 | 20.856 | 0.921 | 0.998 | 30.487 | 0.859 | 0.915 | 3.926 | -0.953 | 0.830 | 2.586 |

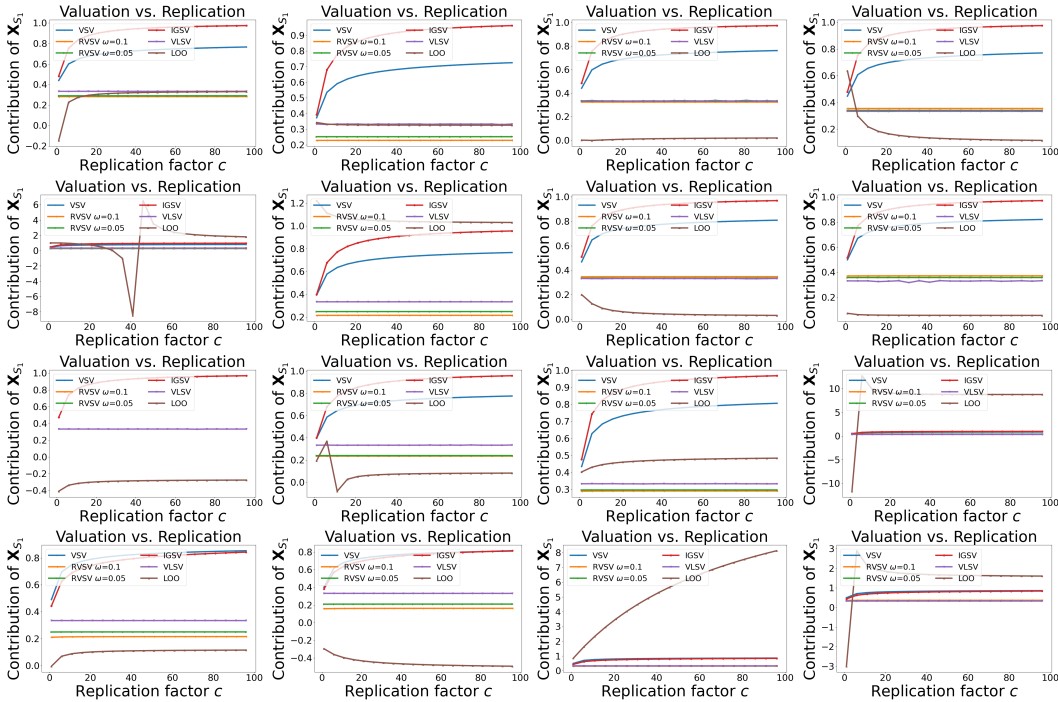

Figure 16: Valuations for the replicated dataset $\mathbf{X}_{S_1}$ under two data distributions: *supersets* of ratios 0.1 & 1 (left two) and *disjoint* of ratios 0 & 1 (right two). The vertical axis shows $\mathbf{X}_{S_1}$'s value. The horizontal axis shows the replication factor $c$. From first row to last row: CaliH, KingH, USCensue, FaceA. Note The disjoint ratio 0 for USCensus dataset leads to $NaN$ values for VSV so we show from disjoint ratio 0.2.

## B.7 IGSV vs. RVSV

**Effect of hyperparameters on the SV**. The setting of the experiments is consistent as described previously, including data distributions and models used. In IGSV, a crucial hyperparameter is the user-specified prior $\sigma$ in the covariance, namely assuming the random variables of interest follow $\mathcal{N}(\mathbf{0}, \sigma^2\mathbf{I})$ [35]. In this case, the random variables are the parameters of the linear regressor. Here we vary the $\sigma \in \{0.0001, 0.001, 0.01, 0.1, 1\}$. For RVSV, we vary the $\omega \in \{0.01, 0.05, 0.1, 0.2, 0.25\}$ and calculate the $\log()$ of RV for numerical stability and due to Lemma 7 below which sheds some light on the similarity between RV and IG. The results on four datasets CaliH, KingH, USCensus and FaceA for the two non-i.i.d. distributions: *supersets* and *disjoint* are in Fig. 17.

We make two observations. 1): For IGSV, different priors lead to different SV with the same $\mathbf{X}_{S_1}, \mathbf{X}_{S_2}, \mathbf{X}_{S_3}$. The data providers will thus want to use the prior which leads to their SV being the highest, and may result in disagreement. For RVSV, all the experimented $\omega$ values coincide with the same SV, avoiding this potential selection over $\omega$. 2): $\ln()$ function is plotted as a growth rate reference. While in some cases IGSV grows slower than $\ln()$, it does not converge, implying that a data provider can always have non-zero additional gain with more replication, especially for $c \leq 20$. Confirming the result of our previous replication experiments that IGSV may be less robust. In contrast, RVSG stays consistent regardless of $c$.

**An interesting connection from volume to information gain**. IGSV leverages the information gain criterion, or alternatively the conditional entropy criterion. The intuitive interpretation is given $\mathbf{X}_{S_1}, \mathbf{X}_{S_2}$ is valuable if $\mathbf{X}_{S_2}$ provides additional and new information that is *not* captured by $\mathbf{X}_{S_1}$. Interestingly, $\log \mathrm{Vol}()$ offers an echoing interpretation via Lemma 7. The intuition is also similar, given $\mathbf{X}_{S_1}, \mathbf{X}_{S_2}$ is valuable if $\mathbf{X}_{S_2}$ "occupies" additional space that is *not* "occupied" by $\mathbf{X}_{S_1}$.

Furthermore, this similarity inspired us to relate to the duality of *maximum entropy sampling* [34], which gives rise to the following practical implication: a greedy iterative approach to maximize the

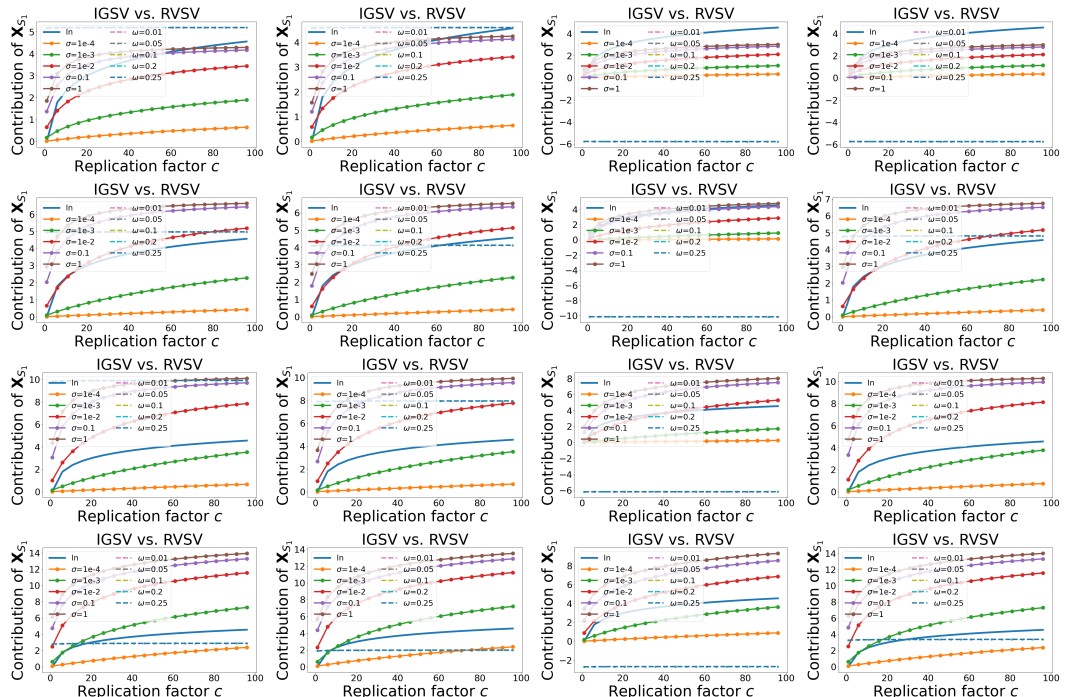

Figure 17: IGSV (solid lines with dot) and RVSV (dashed lines) vs. replication factor $c$. $\sigma$ denotes the prior on the standard deviation for IGSV. $\omega$ is the discretization for RVSV. Valuations for the replicated dataset $\mathbf{X}_{S_1}$ under two data distributions: *supersets* of ratios $0.1$ & $1$ (left two) and *disjoint* of ratios $0$ & $1$ (right two). The $y$-axis shows $\mathbf{X}_{S_1}$'s value. The $x$-axis shows the replication factor $c$. The first to last row: CaliH, KingH, USCensus, FaceA.

$\log \mathrm{Vol}()$ in data collection/purchase gives a near-optimal solution ($(1 - 1/e)$-approximation) in terms maximizing the volume. See detailed discussion below.

**Lemma 7** (**Duality of Volume Decomposition**). *For full-rank* $\mathbf{X}_S, \mathbf{X}_{S'}$ *of* $\mathbf{X}$*, we have*

$$\log V_{S \cup S'} = \log V_S + 0.5 \times \log V_{S|S'}$$

*where* $\log V_{S|S'} := \log |\mathbf{I} + \mathbf{G}_S^{-1} \mathbf{G}_{S'}|$.

**Proof of Lemma 7.**

$$V_{S \cup S'}^2 := |\mathbf{X}^\top \mathbf{X}| = |\mathbf{G}_S + \mathbf{G}_{S'}| = |\mathbf{G}_S \times (\mathbf{I} + \mathbf{G}_S^{-1} \mathbf{G}_{S'})|$$

$$= |\mathbf{G}_S| \times |\mathbf{I} + \mathbf{G}_S^{-1} \mathbf{G}_{S'}| = V_S^2 |\mathbf{I} + \mathbf{G}_S^{-1} \mathbf{G}_{S'}| \quad \text{Taking log on both sides}$$

$$2 \log(V_{S \cup S'}) = 2 \log(V_S) + \log(|\mathbf{I} + \mathbf{G}_S^{-1} \mathbf{G}_{S'}|)$$

$$\log(V_{S \cup S'}) = \log(V_S) + 0.5 \times \log(|\mathbf{I} + \mathbf{G}_S^{-1} \mathbf{G}_{S'}|)$$

$\square$

We explicitly write $V = V_{S \cup S'}$ and define a notation $V_{S|S'}$ to better illustrate the similarity to the duality in *maximum entropy sampling* where maximizing the entropy of the selected set ($\mathbb{H}[\mathbf{f}_\mathcal{O}]$) minimizes the conditional entropy ($\mathbb{H}[\mathbf{f}_{\mathcal{X} \setminus \mathcal{O}} | \mathbf{f}_\mathcal{O}]$) on the remaining set as follows,

$$\mathbb{H}[\mathbf{f}_\mathcal{X}] = \mathbb{H}[\mathbf{f}_\mathcal{O}] + \mathbb{H}[\mathbf{f}_{\mathcal{X} \setminus \mathcal{O}} | \mathbf{f}_\mathcal{O}]$$

where $\mathcal{X}$ denotes the input space for the input locations to be observed, and $\mathcal{O} \subseteq \mathcal{X}$ is a selected subset of the input locations and $\mathbf{f}()$ is the unknown function of interest. $\mathbb{H}(\cdot)$ is the standard differential entropy.

$V_S$ depends on the relation between $\mathbf{X}_S$ and $\mathbf{X}_{S'}$ in how well $\mathbf{X}_S$ covers the space relative to $\mathbf{X}_{S'}$ as reflected in $|\mathbf{I} + \mathbf{G}_{S'} \mathbf{G}_S^{-1}|$. $V_S$ is large if $\mathbf{X}_S$ covers the remaining subset $\mathbf{X}_{S'}$ (implying $V_{S'}$ will be

small). This is because $V_{S \cup S'}$ is a constant. Similarly, in MES, if $\mathbb{H}[\mathbf{f}_\mathcal{O}]$ is large, then $\mathbb{H}[\mathbf{f}_{\mathcal{X} \setminus \mathcal{O}} | \mathbf{f}_\mathcal{O}]$ will be small, because $\mathbb{H}[\mathbf{f}_\mathcal{X}]$ is a constant.

A practical implication is thus: suppose a data provider already knows what data can be collected (denoted by $S \cup S'$ to be consistent with previous discussion), and wants to maximize the value of collected data (denoted by $S$) under a constrained budget (e.g. time, memory, and other processing costs), an iterative greedy approach to update $S, S'$ as follows:

$$\underset{\mathbf{x}_i}{\operatorname{argmax}} \ \log V_S - \log V_{S|S'}$$

$$S \leftarrow S \cup \{\mathbf{x}_i\}$$

yields an $(1 - 1/e)$-approximation [22] if $\log \operatorname{Vol}()$ as in our definition is submodular. Note our proof is different from the examples [26] which define the squared volume to be the determinant of the (right) Gram matrix. That definition admits a simpler proof technique by a geometric argument that is not applicable to our definition of volume.

First, we recall the definition of submodularity. Let $[n]$ denote the set $\{1, \ldots, n\}$ as the indices of the rows/data points in $\mathbf{X}$, there are thus $2^n$ possible subsets of $[n]$. A set function $g : 2^n \mapsto \mathbb{R}$ is called *submodular* if for any $S \subseteq S^+ \subseteq [n]$ and $\forall \mathbf{x} \in \mathbf{X}$ ($\mathbf{x}$ is a data point in $\mathbf{X}$),

$$\underbrace{g(S \cup \{\mathbf{x}\}) - g(S)}_{\Delta_S} \geq \underbrace{g(S^+ \cup \{\mathbf{x}\}) - g(S^+)}_{\Delta_{S+}}.$$

**Proposition 9** (Submodularity of $\log \operatorname{Vol}()$)**.** *If for any $S \subseteq S^+ \subseteq [n]$, $\mathbf{G}^+ - \mathbf{G}$ is positive semi-definite where $\mathbf{G} := \mathbf{X}_S^\top \mathbf{X}_S$ and $\mathbf{G}^+ := \mathbf{X}_{S+}^\top \mathbf{X}_{S+}$, then $\log \operatorname{Vol}()$ is submodular.*

*Proof.* By letting $g() = \log \operatorname{Vol}()$ and a direct application of Lemma 1 , we have: $\Delta_S = 1/2 \times \log(1 + \mathbf{x}(\mathbf{X}_S^\top \mathbf{X}_S)^{-1} \mathbf{x}^\top)$ and $\Delta_{S+} = 1/2 \times \log(1 + \mathbf{x}(\mathbf{X}_{S+}^\top \mathbf{X}_{S+})^{-1} \mathbf{x}^\top)$. We want to show:

$$\Delta_S \geq \Delta_{S+}$$

$$1/2 \times \log(1 + \mathbf{x}(\mathbf{X}_S^\top \mathbf{X}_S)^{-1} \mathbf{x}^\top) \geq 1/2 \times \log(1 + \mathbf{x}(\mathbf{X}_{S+}^\top \mathbf{X}_{S+})^{-1} \mathbf{x}^\top)$$

$$\mathbf{x}(\mathbf{X}_S^\top \mathbf{X}_S)^{-1} \mathbf{x}^\top \geq \mathbf{x}(\mathbf{X}_{S+}^\top \mathbf{X}_{S+})^{-1} \mathbf{x}^\top$$

$$\mathbf{x} \left[ (\mathbf{X}_S^\top \mathbf{X}_S)^{-1} - (\mathbf{X}_{S+}^\top \mathbf{X}_{S+})^{-1} \right] \mathbf{x}^\top \geq 0.$$

Next, by a direct application of Lemma 8 below: substituting $\mathbf{A} = \mathbf{X}_S^\top \mathbf{X}_S, \mathbf{B} = \mathbf{X}_{S+}^\top \mathbf{X}_{S+}$, we can show $(\mathbf{X}_S^\top \mathbf{X}_S)^{-1} - (\mathbf{X}_{S+}^\top \mathbf{X}_{S+})^{-1}$ is positive semi-definite and the proof is complete. $\qquad \square$

The assumption $\mathbf{G}^+ - \mathbf{G}$ is positive semi-definite, has the interpretation that the left Gram of a larger dataset ($\mathbf{X}_{S+}$) has "more" information (the entire result is verified empirically after the proof). Fig. 18 shows the proportion of $\log \operatorname{Vol}()$ is submodular over 500 independent trials of randomly sampled matrices $\mathbf{X}_S$ (and subsequently constructed $\mathbf{X}_{S+}$). We observe $\log \operatorname{Vol}()$ is almost always submodular. The exceptions may be attributed to that randomly drawn matrices may sometimes violate the condition of $\mathbf{G}, \mathbf{G}^+$ being positive definite required by Lemma 1, or equivalently $\mathbf{X}_S$ may not be full-rank if $n$ is small relative to $d$.

While the lemma below is with respect to matrices, if we consider the scalar version, it is much more intuitive: for two positive scalars $a, b > 0, a \geq b \implies 1/b \geq 1/a$. The result essentially generalizes this idea to symmetric and positive definite matrices. We will use $\mathbf{A} \succeq \mathbf{0}$ to denote $\mathbf{A}$ is positive semi-definite.

**Lemma 8.** *Given $\mathbf{A}, \mathbf{B} \in \mathbb{R}^{d \times d}$ are both symmetric and positive definite, then*

$$\mathbf{B} - \mathbf{A} \succeq \mathbf{0} \implies \mathbf{A}^{-1} - \mathbf{B}^{-1} \succeq \mathbf{0}$$

*Proof.*

$$\mathbf{B} - \mathbf{A} \succeq \mathbf{0}$$

$$\implies \mathbf{A}^{-1/2}(\mathbf{B} - \mathbf{A})\mathbf{A}^{-1/2} \succeq \mathbf{0}$$

$$\implies \mathbf{A}^{-1/2}\mathbf{B}\mathbf{A}^{-1/2} \succeq \mathbf{I}$$

$$\implies \mathbf{A}^{1/2}\mathbf{B}^{-1}\mathbf{A}^{1/2} \preceq \mathbf{I},$$

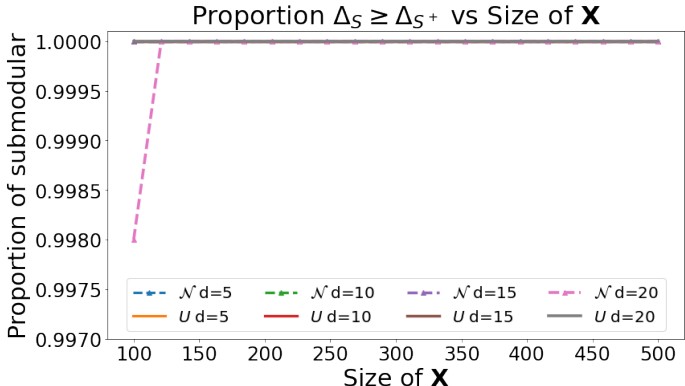

Figure 18: The proportion of $\log \text{Vol}()$ is submodular with respect to randomly drawn $\mathbf{X}_S, \mathbf{X}_{S^+}, S \subset S^+$. $d$ denotes feature dimension. $\mathcal{N}$ denotes $\mathbf{X}$ is sampled from the standard normal distribution $\mathcal{N}(\mathbf{0}, \mathbf{I})$ and $U$ denotes $\mathbf{X}$ is sampled from the uniform distribution $U(0, 1)$. $\mathbf{X}_S$ is then a randomly sampled submatrix of $\mathbf{X}$, containing $0.2 \times n$ number of data points of $\mathbf{X}$. $\mathbf{X}_S^+$ is constructed by appending a randomly select data point from $\mathbf{X}$ to $\mathbf{X}_S$. The proportion is calculated over 500 independent random trials. We observe that $\log \text{Vol}()$ is almost always submodular, except in some degenerate cases where the randomly drawn $\mathbf{X}_S$ is not full-rank.

therefore,

$$\mathbf{B}^{-1} = \mathbf{A}^{-1/2}\mathbf{A}^{1/2}\mathbf{B}^{-1}\mathbf{A}^{1/2}\mathbf{A}^{-1/2}$$
$$\preceq \mathbf{A}^{-1/2}\mathbf{I}\mathbf{A}^{-1/2}$$
$$\preceq \mathbf{A}^{-1}.$$

The first implication is by using the fact that $\mathbf{B} - \mathbf{A} \succeq \mathbf{0} \implies \mathbf{C}^\top\mathbf{A}\mathbf{C} \preceq \mathbf{C}^\top\mathbf{B}\mathbf{C}$ for any conformable matrix $\mathbf{C}$ and viewing $\mathbf{A}^{-1/2}$ as a conformable matrix. The second implication is by considering the relationship between a symmetric and positive definite matrix $\mathbf{B}$ and $\mathbf{I}$, where $\mathbf{I} \preceq \mathbf{B} \implies \mathbf{B}^{-1} \preceq \mathbf{I}$. The following steps are substitutions and applications of the definition of $\succeq$. $\qquad \square$