# OpenReview forum: "Validation Free and Replication Robust Volume-based Data Valuation"
_NeurIPS.cc/2021/Conference — NeurIPS 2021 Poster_

### Official Review · Reviewer_WiPz · 2021-07-14

**Rating:** 7
**Confidence:** 4

**Summary:**

The paper proposes a measure of datasets’ values using the volume of the feature matrix (defined to be the determinant of its left Gram), which captures the diversity in features. They show the effectiveness of this measure both theoretically and empirically. The paper proves that for regression problems,  a larger volume usually leads to a smaller bias and a lower mean squared error. They also verify this claim using randomly generated datasets. They also proposed a replication robust measure by compressing similar data. Finally, they use experiments on both real-world datasets and synthetic data to verify their claims and demonstrate that their measures without validation give results consistent with other methods which may require validation.

**Limitations And Societal Impact:**

Yes.

**Main Review:**

The paper proposes a novel data valuation method that does not require validation using test data, which can also be robust to data replication. The paper is strong in the sense that it provides both solid theoretical guarantees and extensive empirical evaluations of their measures. I find no obvious weakness of the paper.

The paper is well written and easy to follow. I didn’t find any glaring mistakes in the results and the proofs.

Some other relevant papers:
“A Distributional Framework For Data Valuation”  Amirata Ghorbani, Michael Kim, James Zou
“Truthful Data Acquisition via Peer Prediction” Yiling Chen, Yiheng Shen, Shuran Zheng
“Replication-Robust Payoff-Allocation for Machine Learning Data Markets” Dongge Han, Michael Wooldridge, Alex Rogers, Shruti Tople, Olga Ohrimenko, Sebastian Tschiatschek



**Time Spent Reviewing:**

4

---

> ### Author Response · Authors · 2021-08-10
> **Author Response**
>
> We thank the reviewer for taking the time to review our paper and the positive feedback. Moreover, we thank the reviewer for bringing relevant literature to our attention, we will carefully consider and discuss them where appropriate.

---

> ### Author Response · Authors · 2021-08-29
> **Thanks to reviewer WiPz**
>
> We would like to thank the reviewer for taking the time to review our paper and for recognizing the novelty of our approach and both the theoretical and empirical contributions. Kindly let us know if you might have further comments and we will do our best to address them in the remaining time.

---

> ### Comment · Reviewer_WiPz · 2021-08-29
> **Thanks for the reply**
>
> Thanks for the reply. I have no further questions about the paper.

---

### Official Review · Reviewer_deSj · 2021-07-16

**Rating:** 6
**Confidence:** 5

**Summary:**

The paper proposes to use the volume of the data matrix as a data value metric. The contributions of the paper include (1) presenting theoretical and empirical evidence about the connection between volume and learning performance (2) designing a robust volume and (3) experimental validation

**Limitations And Societal Impact:**

The limitations of the paper is not sufficiently discussed. For the feedbacks, please refer to the previous section.

**Main Review:**

Pros:
- the paper is well-written
- the theoretical analysis of the paper is solid

Cons:
- While the reviewer appreciates the solid theoretical analysis around volume and replication-robust volume, the motivations for this paper are questionable. One of the motivations is to have a validation set-free data value. In practice, the data's value is about  not only performance but other ethical concerns like fairness. With a validation set, it's flexible to encode fairness by maybe using an unbiased validation set. Another example is when the data buyer wants to build a model for a targeted population; then the validation set could include only data from a certain population. With a validation set, it's convenient and flexible to encode data buyer's expectations about data, which may vary from application to application. However, the volume does not seems to offer this flexibility. The second motivation is to enable replication robustness. Although this is interesting theoretically, there are very mature existing deduplication techniques in the database literature. In practice, data replicates would not be much of a concern because one can just run these duplication techniques and then perform data valuation.
- The use case of volume is limited to linear regression thus far; it's not clear how to extend it to other widely-used models.


**Time Spent Reviewing:**

2

---

> ### Author Response · Authors · 2021-08-10
> **Author Response**
>
> We thank the reviewer for taking the time to review our paper and appreciating our theoretical analysis.
>
> We would like to address the concerns as follows:
> - In the case where a suitable validation set for evaluating dataset values is readily available, we agree with the reviewer that using this validation set is an effective approach. However, we wish to highlight that __it is often challenging to obtain such a suitable validation set__. Therefore, __our approach can be particularly useful in scenarios where the validation set is not available__ because our approach is based on examining the inherent property (i.e., diversity) of the data, formalized by the volume. We wish to highlight several practical challenges in obtaining a suitable validation set: i) additional cost to collect high-quality data. The cost is reflected in terms of the expensive and time-consuming data collection and labeling process [4], especially when the data will be used to assess the performance of trained models; ii) a disagreement on the choice of the validation set may arise [1]. In particular, __we empirically demonstrate how such an impeding disagreement could arise__ (Sec.5.2, Fig.5, lines 302-313) __on a real-world credit card transaction dataset__. Specifically, we emulate the case where smaller banks prefer transaction data with lower amounts (<\\$1000) and larger banks prefer transaction data with higher amounts (>\\$1000). The validation-based approach (i.e., VLSV) gives different values for the same dataset. In such cases, it is not immediately clear how to select a suitable validation set. In contrast, our approach stems from the intuition that “a more diverse dataset gives better learning performance” (in lines 132-134). __Relying on the inherent diversity to measure dataset values, our method can be useful when a suitable validation set is unavailable__ due to the aforementioned issues.
>
> - We wish to clarify two points.
> The first is that we also consider __noisy replication where data are duplicated first and perturbed with a small amount of noise__, in lines 229-230 (also in [3]). While it is unclear how a standard deduplication technique would be effective in this scenario, __our method is empirically demonstrated to have some degree of robustness__ in the experimental results in Appendix B.2 and B.3.
> The second point is that __we formalized a simple deduplication technique in Proposition 7 and provided a comparative discussion__ in lines 227-230. Specifically, we refer to it as $\text{RV}_{\mathbb{1}}(\cdot;\omega)$. It has a specific disadvantage that it might remove multiple appearances of honest data which are representative of the true distribution. For example, suppose in the true distribution some support region has high density, then it is possible multiple same data are sampled from this support, and their multiple appearances should be carefully considered instead of simply removed as in standard deduplication techniques [2]. It is important in machine learning to carefully consider the multiple appearances as they help to characterize the true distribution. Our approach can achieve this via a hyperparameter $\alpha$, see the definition of $\rho_i$ in Equ.(2) of Definition 3 in line 188. In short, having a larger $\alpha$ increases the value of multiple appearances while having a smaller $\alpha$ does the opposite. We also describe an intuitive guide on how to set $\alpha$ (in lines 209 - 216) and formalize this intuition with theoretical results (Proposition.8 and Lemma.5) in Appendix A.3. We wish to further highlight that deduplication techniques may not be effective in addressing replication and preserving the possible density information about the true distribution simultaneously, while both are important in the context of machine learning.
>
> - We wish to point out that, __not restricted to linear regression, our empirical investigation has explored beyond our theoretical results to demonstrate the effectiveness of our method on more complex models__. Specifically, in Sec.5.3 we investigate several such extensions:
>     - Lines 317-320: we adopt the __GloVe word embedding__ and construct a __Bidirectional LSTM__ with a fully connected layer of 8 hidden units on a real-world text reviews dataset.
>     - Lines 339-340 of main text & lines 765-767 in Appendix B.6: we adopt __deep neural networks of 2 hidden layers__ on three real-world datasets.
>     - Lines 767-770 in Appendix B.6: we adopt a __3-layer convolutional neural network with batch normalization__ on a real-world image dataset.
>
>     In the above experiments on more complex models, we have demonstrated promising results in terms of providing consistent valuation with existing baselines when there is no replication (Sec. 5.2); and showing replication robustness (Sec.5.3).
>
>
> - We wish to address the comment regarding limitations by highlighting two parts in our paper, specifically the trade-offs introduced by hyper-parameters (Sec.4.2 lines 209-228) and the necessary assumption to use standardized features (lines 235 - 240).
>
>     We discuss the trade-offs via the introduced hyper-parameters $\alpha$ and $\omega$. For $\alpha$, we provide an intuitive discussion on the selection in lines 209 - 216 and formalize it via Proposition 8 and Lemma 5 in Appendix A.3. For $\omega$, we discuss the diversity representation vs. robustness trade-off in lines 219-222 and formalize it with the ensuing Proposition 7. We in addition include a comparative discussion to highlight the contrasting effects in balancing this trade-off by explicitly generalizing our method to the two extremes: $\text{RV}_{\mathbb{1}}(\cdot;\omega)$ and $\text{Vol}()$ in lines 227-228. This highlights the flexibility of our proposed method in the said trade-off.
>
>     In line 235, “we assume the features follow a normal distribution”. In lines 236-237: we provide an intuitive interpretation of this assumption “data far away from the mean are more useful for learning, because they are statistically rarer, so we assign them with higher values.” Furthermore, other existing methods in our empirical investigation also confirm this interpretation, as shown in Sec.5.2, lines 295-299.
>
> We hope our clarifications could improve your opinion of our paper.
>
> #### References:
> [1] R. H. L. Sim, Y. Zhang, M. C. Chan, and B. K. H. Low. Collaborative machine learning with incentive-aware model rewards. In Proc. ICML, pages 8927–8936, 2020
>
> [2] Wen Xia, Hong Jiang, Dan Feng, Fred Douglis, Philip Shilane, Yu Hua, Min Fu, Yucheng Zhang, and Yukun Zhou. A Comprehensive Study of the Past, Present, and Future of Data Deduplication. In Proceedings of the IEEE, vol. 104, no. 9, pages 1681-1710, 2016.
>
> [3] Dongge Han, Michael Wooldridge, Alex Rogers, Shruti Tople, Olga Ohrimenko, and Sebastian Tschiatschek. Replication-robust payoff-allocation for machine learning data markets. arXiv:2006.14583, 2021
>
> [4] Y. Roh, G. Heo and S. E. Whang, A Survey on Data Collection for Machine Learning: A Big Data - AI Integration Perspective. In IEEE Transactions on Knowledge and Data Engineering, vol. 33, no. 4, pp. 1328-1347, 2021

---

> ### Comment · Reviewer_deSj · 2021-08-29
> **post rebuttal**
>
> Thanks for the authors' detailed response.
>
> >> it is often challenging to obtain such a suitable validation set.
>
> In practice, given the training data, the validation set is often taken as part of training data.
>
> >>  ii) a disagreement on the choice of the validation set may arise [1]. In particular, we empirically demonstrate how such an impeding disagreement could arise (Sec.5.2, Fig.5, lines 302-313) on a real-world credit card transaction dataset. Specifically, we emulate the case where smaller banks prefer transaction data with lower amounts (<$1000) and larger banks prefer transaction data with higher amounts (>$1000).
>
> I would not treat this as a disadvantage of the validation set-based data valuation, but an advantage. If the model user cares about predicting well on low transactions, then the validation set should be chosen accordingly and the data valuation algorithm should prefer points that contribute to improving prediction on low transactions. Data values depend on how the data is used; with different prediction goals, data values should naturally be different. The current approach, although technically sound, does not fully respect the context-dependent nature of data value.
>
> My other concerns are addressed. I would raise my score to 6.

---

> > ### Author Response · Authors · 2021-08-29
> > **Thanks to reviewer deSj**
> >
> > We would like to thank the reviewer for taking the time to review our paper and for appreciating our theoretical analysis and in particular for raising the interesting discussion on the role of validation in data valuation. We agree with the reviewer that context-dependent valuation using  _a suitable validation set_ is an effective approach. We hope our work provides a useful validation-free perspective for data valuation when the validation is unavailable, e.g. a model user wants to predict well on low transactions but does not actually have labeled data to conduct validation (which is why the user wants to buy data in the first place). Kindly let us know if you have further questions and we will do our best to address them in the remaining time.

---

### Official Review · Reviewer_s1Fn · 2021-08-04

**Rating:** 6
**Confidence:** 3

**Summary:**

This paper intends to design a method to evaluate the value of a dataset. Such methods have been previously developed by calculating the Shapley value of the dataset based on its effect on model accuracy. The authors argued that since the model accuracy depends on a validation data set, selection of the validation dataset highly affects the Shapley value of each party's dataset, and hence will be a source of disagreement. To overcome this issue the authors set the value of a dataset based on the determinant (volume) of the left Gram matrix of its input features. The value of duplicated data is removed by constructing a compressed version of the dataset that preserves its diversity. The authors refer to this as robustness, since the values are robust to duplications. The authors provide some propositions connecting volume with bias and MSE. This has been validated experimentally as well.


**Ethical Concerns:**

There is no ethical concern

**Limitations And Societal Impact:**

The authors did not discuss any societal impact. I don't think there is any major negative ones.

**Main Review:**

This paper intends to design a method to evaluate the value of a dataset. Such methods have been previously developed by calculating the Shapley value of the dataset based on its effect on model accuracy. The authors argued that since the model accuracy depends on a validation data set, selection of the validation dataset highly affects the Shapley value of each party's dataset, and hence will be a source of disagreement. To overcome this issue the authors set the value of a dataset based on the determinant (volume) of the left Gram matrix of its input features. The value of duplicated data is removed by constructing a compressed version of the dataset that preserves its diversity. The authors refer to this as robustness, since the values are robust to duplications. The authors provide some propositions connecting volume with bias and MSE. This has been validated experimentally as well.

It is nice that the authors decoupled valuation and validation. However, their approach comes with its own issues. For example, in reality, there is a higher value to a point that appears multiple times in a dataset. (To understand this think of an extreme case where the points are uniform samples of the true distribution and 99% of them are the same point.) But this approach set the value of the duplicates to 0. If removing the value of the duplicates is because we don't trust that the parties are going to share their actual data point, they may as well generate completely fake datapoints.

20: ... such as the gold standard for oil. Does not read well?
44:  to better performance ->  to a better performance


**Time Spent Reviewing:**

2

---

> ### Author Response · Authors · 2021-08-10
> **Author Response**
>
> ### Author response
>
> We thank the reviewer for taking the time to review our paper and for recognizing our effort in decoupling valuation from validation.
>
> We would like to address the concerns as follows:
> - We wish to point out that __our approach does not set the value of the duplicates to 0__. Instead, we carefully control the weight of multiple appearances of the same data point using a hyperparameter $\alpha$, see the definition of $\rho_i$ in Equ.(2) of Definition 3 in line 188. In short, __having a larger $\alpha$ increases the value of multiple appearances while having a smaller $\alpha$ decreases this value__. We also give an intuitive guide on how to set $\alpha$ (in lines 209 - 216) and formalize this intuition with theoretical results (Proposition.8 and Lemma.5) in Appendix A.3. In addition, we also specifically discuss the disadvantage of setting the value of duplicate to 0 (in lines 227-228) where we formalized it as $\text{RV}_{\mathbb{1}}(\cdot;\omega)$ and point out its disadvantage. For example, suppose in the true distribution some support region has high density, then it is possible multiple same data are sampled from this support, and their multiple appearances should be carefully considered instead of simply removed. It is important in machine learning to carefully consider the multiple appearances as they help to characterize the true distribution and our method can achieve this by carefully setting $\alpha$.
>
> - Regarding the comment on “fake datapoints”, we wish to point out that in lines 238-240, “we restrict our discussion to exclude outliers as they are not truly representative of the actual distribution”, which by definition also excludes such consideration. This setting is motivated by use cases where the data providers are hospitals and/or healthcare organizations [2,3,4,5] and we believe it is unlikely that such organizations would fake medical records for profit. However, duplicates are possible for patients who seek second opinions and undergo repeated tests. This is also a reason we do not set the values of the duplicates to 0 but carefully control their values via $\alpha$.
>
>     Importantly, __our explored setting generalizes that in [1] to additionally consider and effectively address the issue of replication [5,6]__. Both theoretical results (Proposition 6.) and empirical evidence (in Sec.5.3 and Appendix B.7) demonstrate the effectiveness of our proposed method as compared to [1]. We wish to point out that this line of work is relatively new, hence we believe the issues being addressed in our paper (we establish theoretical connections between volume, learning performance and the value of data, and empirically demonstrate this on real-world datasets; we address the issue that the original volume definition is not replication-robust in a principled way with theoretical results and empirical evidence on real-world datasets) are significant and addressing all the issues including trust in the providers simultaneously is challenging. We hope our work would serve as an inspiration to more related works on these issues, such as trust in the data providers.
>
> We thank the reviewer for pointing out typos and phrasing, and will carefully improve our writing and correct the typo.
>
> We hope our clarifications would improve your opinion of our work.
>
>
> #### References
> [1] R. H. L. Sim, Y. Zhang, M. C. Chan, and B. K. H. Low. Collaborative machine learning with incentive-aware model rewards. In Proc. ICML, pages 8927–8936, 2020
>
> [2] Center for Open Data Enterprise. Sharing and utilizing health data for AI applications. Roundtable report, 2019.
>
> [3] Weinstein, M., & Skinner, J. (2011). Data Sharing from Clinical Trials — A Research Funder’s Perspective. New England Journal of Medicine, 362(5), 567–571.
>
> [4] Jackevicius, C. A., An, J., Ko, D. T., Ross, J. S., Angraal, S., Wallach, J. D., Koh, M., Song, J., & Krumholz, H. M. (2019). Submissions from the SPRINT Data Analysis Challenge on clinical risk prediction: A cross-sectional evaluation. BMJ Open, 9(3), 1–9.
>
> [5] Drazen, J. M., Morrissey, S., Malina, D., Hamel, M. B., & Campion, E. W. (2016). The importance - And the complexities - Of data sharing. New England Journal of Medicine, 375(12), 1182–1183.
>
> [6] Han, D., Wooldridge, M., Tople, S., Ohrimenko, O., Rogers, A., & Tschiatschek, S. (2020). Replication-Robust Payoff-Allocation with Applications in Machine Learning Marketplaces. ArXiv.

---

> ### Author Response · Authors · 2021-08-29
> **Thanks to reviewer s1Fn**
>
> We would like to thank the reviewer for taking the time to review our paper and for recognizing our effort in decoupling valuation and validation, and in particular for verifying our theoretical contributions and empirical results. Kindly let us know if our response to your comments on dealing with repeated appearances of the same data point is unclear and we can do our best to clarify in the remaining time.

---

### Decision · Program_Chairs · 2021-09-27

**Decision:**

Accept (Poster)

**Comment:**

The paper studies an important area. The reviewers had some concern about the motivation for validation-free data value and replication robustness, and limitations of the model studies in this paper. Still, the general agreement was that validation-free data value would be important when there are not enough test data, and while the method has its limitations, it's still a worthwhile contributions. Overall, while a borderline accept, the votes lean toward accepting the paper.